# TRABID inhibition activates cGAS/STING-mediated anti-tumor immunity through mitosis and autophagy dysregulation

Yu-Hsuan Chen [1,7], Han-Hsiun Chen [1,7], Won-Jing Wang[2], Hsin-Yi Chen[3], Wei-Syun Huang[2], Chien-Han Kao[1,2], Sin-Rong Lee[1,4], Nai Yang Yeat [1,5,6], Ruei-Liang Yan[1], Shu-Jou Chan[1], Kuen-Phon Wu [1,4] & Ruey-Hwa Chen [1,4] ✉

Activation of tumor-intrinsic innate immunity has been a major strategy for improving immunotherapy. Previously, we reported an autophagy-promoting function of the deubiquitinating enzyme TRABID. Here, we identify a critical role of TRABID in suppressing anti-tumor immunity. Mechanistically, TRABID is upregulated in mitosis and governs mitotic cell division by removing K29-linked polyubiquitin chain from Aurora B and Survivin, thereby stabilizing the entire chromosomal passenger complex. TRABID inhibition causes micronuclei through a combinatory defect in mitosis and autophagy and protects cGAS from autophagic degradation, thereby activating the cGAS/STING innate immunity pathway. Genetic or pharmacological inhibition of TRABID promotes anti-tumor immune surveillance and sensitizes tumors to anti-PD-1 therapy in preclinical cancer models in male mice. Clinically, TRABID expression in most solid cancer types correlates inversely with an interferon signature and infiltration of anti-tumor immune cells. Our study identifies a suppressive role of tumor-intrinsic TRABID in anti-tumor immunity and highlights TRABID as a promising target for sensitizing solid tumors to immunotherapy.

Immune checkpoint blockades (ICBs) targeting PD-1/PD-L1 or CTLA4 have revolutionized the current cancer therapy paradigm based on their high durability[1–3]. However, only a small subset of cancer patients shows a response. Thus, there is an unmet need to fully understand the ICB resistance mechanisms for developing therapeutic strategies that could enhance the response rate to ICBs. Accumulating evidence has revealed that tumors with a low amount of lymphocyte infiltration (so-called cold tumors), a high amount of immunosuppressive factors, or a low tumor mutational burden are often resistant to ICBs[2,4]. In addition, innate cellular immunity is another factor regulating tumor immune microenvironment and determining tumor response to ICBs[5]. Thus,

drugs that elicit tumor-intrinsic innate immunity could not only induce an anti-tumor effect but potentiate tumor response to ICBs.

One prominent innate immunity pathway is the cGAS/STING pathway, in which cGAS senses cytosolic double-stranded DNA to produce cGAMP. cGAMP binds and activates STING to result in the recruitment and activation of TBK1, which in turn phosphorylates IRF3 to facilitate its nuclear translocation for driving the transcription of type I interferons. In addition, STING activates IKK to induce the transcription of NFκB-driven inflammatory genes[6–8]. Given the crucial role of cGAS/STING signaling in the induction of anti-cancer immunity, STING agonists hold a promise to enhance the ICB response for

[1]Institute of Biological Chemistry, Academia Sinica, Taipei 115, Taiwan. [2]Institute of Biochemistry and Molecular Biology, National Yang Ming Chiao Tung University, Taipei 112, Taiwan. [3]Graduate Institute of Cancer Biology and Drug Discovery, College of Medical Science and Technology, Taipei Medical University, Taipei 110, Taiwan. [4]Institute of Biochemical Sciences, College of Life Science, National Taiwan University, Taipei 106, Taiwan. [5]Chemical Biology and Molecular Biophysics Program, Taiwan International Graduate Program, Academia Sinica, Taipei 115, Taiwan. [6]Department of Chemistry, National Tsing Hua University, Hsinchu 300, Taiwan. [7]These authors contributed equally: Yu-Hsuan Chen, Han-Hsiun Chen. ✉e-mail: rhchen@gate.sinica.edu.tw

treating advanced cancer patients. To date, many natural and synthetic STING agonists have been tested in preclinical models and some have entered early clinical trials[9–11]. Although certain promising results have been obtained, the application of STING agonists to clinics remains challenging with the potential pro-tumor functions of STING pathway[12–15] and the difficulty for achieving a persistent STING activation due to the existence of a negative feedback regulation mediated by protein degradation[16,17]. Thus, the development of alternative therapeutic strategies may be needed to overcome the potential limitations of STING agonists.

Since the DNA-sensing ability of cGAS is influenced by chromatin context, rather than DNA source and sequence[18,19], self DNAs accumulated in the cytoplasm enable the activation of cGAS/STING pathway. One important source of cytosolic self-DNAs is the micronuclei arising from defective DNA replication, repair, or mitosis[20–23]. Mitotic defects such as chromosome segregation errors and spindle checkpoint failure induce chromosome instability (CIN) and aneuploidy, which are common characteristics of cancer cells. The chromosomal passenger complex (CPC), consisting of Aurora B, INCENP, Borealin, and Survivin, is targeted to specific locations at different stages of mitosis to govern a series of mitotic events, such as kinetochore assembly, chromosome bi-orientation, spindle assembly checkpoint activation, cytokinesis, and mitotic exit[24,25]. These diverse functions are mainly mediated by Aurora B-dependent phosphorylation of various mitotic regulators. Accordingly, the expression of Aurora B is tightly controlled during the cell cycle through multiple transcriptional and posttranslational mechanisms, including ubiquitination[26,27]. For instance, APC$^{Cdh1}$ is the major E3 ligase responsible for Aurora B degradation at mitotic exit[28,29]. Additionally, the F box proteins FBXW7 and FBXL promote Aurora B ubiquitination and degradation by acting as substrate adaptors of the Cullin 1 ubiquitin ligase[30,31]. Conversely, deubiquitinating enzymes (DUBs) USP35 and USP13 stabilize Aurora B to ensure a proper mitotic progression[32,33]. Besides Aurora B, the CPC subunit Survivin also undergoes ubiquitin-dependent proteolysis and shares the regulatory E3 ligase APC$^{Cdh1}$ and DUB USP35 with Aurora B[34,35].

TRABID, encoded by the *ZRANB1* gene, is a DUB that specifically disassembles K29- and K33-linked ubiquitin chains[36]. We previously identified TRABID as a positive regulator of autophagy by deubiquitinating and stabilizing VPS34, a key autophagic protein[37]. Furthermore, several studies reported the tumor-promoting functions of TRABID. For instance, TRABID is highly expressed in hepatocellular carcinoma (HCC) and promotes HCC proliferation and metastasis through deubiquitinating SP1[38]. In triple-negative breast cancer, TRABID is associated with poor prognosis and facilitates EZH2 stabilization to promote tumor growth and metastasis[39]. Furthermore, TRABID is upregulated in colorectal cancer, correlates with poor prognosis, and promotes cancer stem cell-like features by stabilizing Sox9[40]. However, these pro-tumor effects are related to the enhancement of malignant features of tumor cells. Whether TRABID in tumor cells could impact on tumor microenvironment (TME) remains elusive. In this study, we identify a key role of TRABID in mitotic cell division by deubiquitinating Aurora B and Survivin to result in the stabilization of the entire CPC. Furthermore, TRABID inhibition impairs mitosis and autophagy to boost micronuclei accumulation, thereby activating the cGAS/STING pathway. We provide evidence indicating TRABID as a promising target for inducing anti-tumor immunity and enhancing tumor response to ICBs.

## Results

### TRABID is required for mitotic cell division

In an attempt to establish Trabid knockout (KO) mouse embryonic fibroblasts (MEFs), we infected MEFs derived from *Zranb1* floxed embryos with adenovirus carrying Cre recombinase (AdCre). As a control, MEFs were infected with adenovirus-carrying luciferase (AdLuc). The resulting cells were designated as Trabid KO and control MEFs, respectively. RT-PCR and Western blot analyzes confirmed the elimination of *Trabid* mRNA and protein expression in KO cells at two days after infection (Supplementary Fig. 1a, b). Phase-contrast microscopy revealed morphological abnormalities of Trabid KO MEFs, such as micronuclei and enlarged nuclear areas (Supplementary Fig. 1c–e). These abnormalities were not seen in control or Atg5 KO MEFs, suggesting the existence of an autophagy-independent function of TRABID. Since micronuclei and nuclear area changes are associated with chromosome instability and chromosome number alterations, respectively, our finding suggests a role of Trabid in mitotic cell division. We, therefore, undertook a detailed time-lapse microscopic analysis using H2B-mCherry-expressing Trabid KO and control MEFs to examine the effects of Trabid ablation on mitotic cell division (Fig. 1a and Supplementary Movies 1–3). By monitoring mitotic timing, defined as the time from nuclear envelop breakdown (NEBD) to mitotic exit, we found that mitotic timing was not significantly different between Trabid KO and control cells (Supplementary Fig. 1f). However, the duration from metaphase to mitotic exit was longer in Trabid KO than control MEFs, whereas the duration from NEBD to metaphase was not different (Fig. 1b and Supplementary Fig. 1g). Furthermore, Trabid KO MEFs displayed multiple mitotic defects, such as lagging chromosomes, mis-segregated chromosomes, and chromosome bridges (Fig. 1c–e and Supplementary Fig. 1h). Cumulatively, 54.6% of the Trabid KO MEFs exhibited at least one mitotic defect, comparing to 14.0% observed in the control cells (Fig. 1f). Additionally, a higher percentage of Trabid KO MEFs exhibited cytokinesis failure, compared with control MEFs (Fig. 1g). These findings suggest a crucial role of Trabid in mitotic cell division.

To demonstrate the general role of TRABID in mitotic cell division, we tested whether this function could be recapitulated in human cells. Using HeLa cells stably expressing control or TRABID shRNAs together with H2B-mCherry (Supplementary Fig. 1i), we found that TRABID depletion in HeLa cells similarly impaired mitotic progression and increased mitotic errors (Fig. 1h, Supplementary Fig. 1j, and Supplementary Movies 4 and 5). Transient knockdown of TRABID in HeLa cells by siRNA also elevated mitotic errors, which were abrogated by re-expressing wild-type TRABID, but not TRABID C443S/ΔNZF1 mutant (a catalysis- and ubiquitin chain binding-deficient mutant[41]; termed ΔN/CS thereafter) (Fig. 1i, j). Together, our study identifies a crucial role of TRABID in controlling proper mitotic cell division in both human and mouse cells and a requirement of TRABID catalytic and ubiquitin chain-binding activities for this function.

### TRABID is enriched in M phase for binding and upregulating CPC

Next, we determined the molecular mechanism by which TRABID governs mitotic cell division. We hypothesized that TRABID may stabilize key mitosis-governing proteins through deubiquitination. To identify such proteins, we performed a quantitative liquid chromatography-tandem mass spectrometry (LC-MS/MS) analysis to detect protein expression alterations between control and TRABID-depleted HeLa cells (Fig. 2a). With the criteria of fold change (shTRABID/control) < 0.666 and P value < 0.05, 441 proteins were recovered as the downregulated proteins by TRABID depletion from three biological repeats (Supplementary Fig. 2a). To identify mitotic regulators in this pool, we intersected these 441 proteins with 238 mitotic proteins listed in the "mitotic cell cycle" category from both Gene Ontology (GO) and Reactome pathway analytic tools. Five overlapping proteins were uncovered. Among them, INCENP was of the second highest downregulation score (Fig. 2b). Importantly, two other CPC components, Aurora B and Survivin (also known as BIRC5), were also significantly downregulated by TRABID knockdown, albeit with fold changes slightly higher than 0.666. Furthermore, we found that the mRNA and protein levels of TRABID were elevated in mitotically

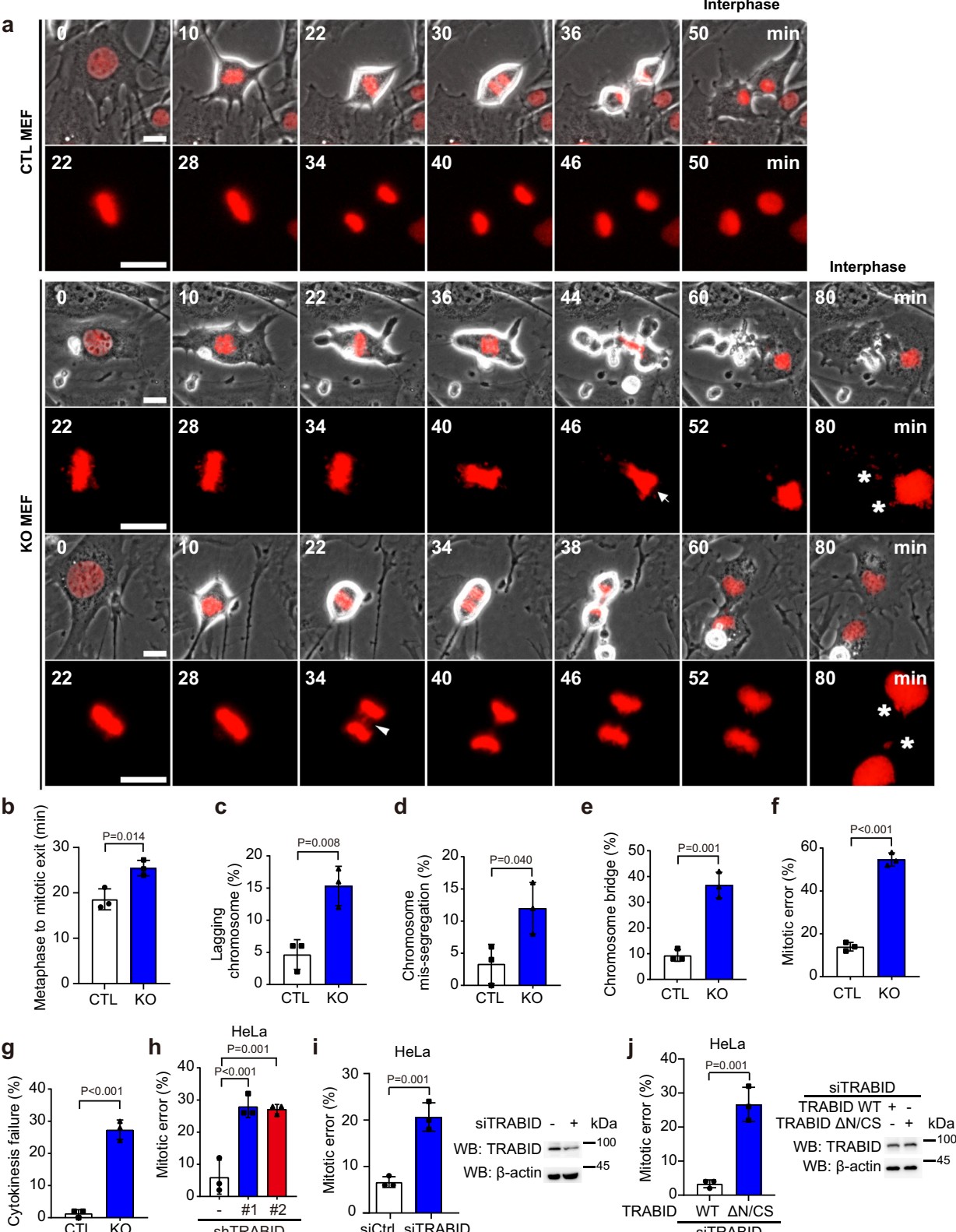

arrested HeLa cells, compared with asynchronized cells or cells arrested in other phases of cell cycle (Fig. 2c, d and Supplementary Fig. 2b). Therefore, we deduced that the effect of TRABID would be more pronounced in mitosis than interphase. Indeed, TRABID knockdown in mitotically arrested HeLa cells led to the downregulation of all CPC components, including Aurora B, Survivin, INCENP, and Borealin (Fig. 2e). Re-expression of TRABID wild type, but not TRABIDΔN/CS

mutant, in TRABID knockdown cells rescued the expression of CPC components (Fig. 2f). Accordingly, TRABID inactivation by treatment of HeLa cells with an inhibitor NSC112200[42] or TRABID knockdown in a murine melanoma cell line B16F10 similarly downregulated CPC components (Supplementary Fig. 2c, d). In line with the reduction of CPC abundance, TRABID knockdown in HeLa cells decreased the phosphorylation of histone H3S10 (Fig. 2g), a known substrate of

**Fig. 1 | TRABID is required for proper mitotic cell division. a** Control (CTL) and Trabid KO (KO) MEFs stably expressing H2B-mCherry (to mark the chromosomes) were examined for mitotic cell division by time-lapse microscopy. Images were taken at 2 min intervals between acquisitions. Time (min) is given relative to the first time frame in late G2 phase. Mitotic errors such as chromosome bridge (arrowhead) and chromosome mis-segregation (arrow) were observed. Micronuclei in the interphase cells are marked by asterisks. Bars, 20 μm. **b** The duration from metaphase to mitotic exit (defined by cell re-attachment) of indicated cell types was counted. Data are mean ± SD ($n = 3$ independent experiments and >50 cells per group per experiment were counted). Each color represents an individual experiment. *P* values are determined by two-sided Student's t-test. **c–g** The percentages of mitotic cells showing lagging chromosomes (**c**), mis-segregated chromosomes (**d**), chromosome bridges (**e**), mitotic errors (**f**), and cytokinesis failure (**g**) were counted. Data are mean ± SD ($n = 3$ independent experiments and >50 cells per group per experiment were counted). *P* values are determined by two-sided Student's t-test. **h** Mitotic errors observed from HeLa cells stably expressing control or TRABID shRNA together with H2B-mCherry. TRABID expression levels and representative time-lapse images are shown in Supplementary Fig. 1i, j, respectively. Data are mean ± SD ($n = 3$ independent experiments and >50 cells per group per experiment were counted). P values are determined by one-way ANOVA with Tukey's post hoc test. **i, j** Mitotic errors observed from HeLa cells stably expressing H2B-mCherry and transfected with TRABID siRNA (**i**) or TRABID siRNA together with TRABID wild type or ΔN/CS construct (**j**). Data are mean ± SD ($n = 3$ independent experiments and >50 cells per group per experiment were counted). *P* values are determined by two-sided Student's t-test. The expression levels of TRABID and mutant are shown. Blots in (**i**), (**j**) are representatives of three independent experiments. Source data are provided as a Source Data file.

Aurora B[43]. Furthermore, reciprocal immunoprecipitation analyzes revealed the interaction between endogenous TRABID and endogenous CPC components in mitotic HeLa cells (Fig. 2h). Proximity ligation assay (PLA) also confirmed the interaction of TRABID with Aurora B in mitotic HeLa cells (Fig. 2i). These findings indicate that TRABID expression is elevated in M phase to act on CPC, thereby contributing to the upregulation of CPC components.

## TRABID deubiquitinates and stabilizes Aurora B and Survivin

Next, we investigated whether CPC subunits are substrates of TRABID. TRABID overexpression diminished the polyubiquitination of Aurora B and Survivin, whereas TRABIDΔN/CS mutant failed to do so (Fig. 3a, b). TRABID overexpression, however, did not affect the ubiquitination of INCENP and Borealin, and the former displayed a low ubiquitination level in cells (Supplementary Fig. 3a, b). In the reciprocal set of experiments, TRABID knockdown in mitotic cells elevated the ubiquitination levels of Aurora B and Survivin (Fig. 3c, d). TRABID is known to target the K29- or K33-linked ubiquitin chains[36] and K29 chains are much more abundant than K33 chains in cells[44,45]. We, therefore, examined the effect of TRABID on K29-ubiquitination of Aurora B and Survivin. Using a synthetic antigen fragment that specifically recognizes the K29-linked ubiquitin chain (termed sAB-K29)[46], we found that overexpression of TRABID, but not TRABIDΔN/CS, greatly reduced K29-polyubiquitination of Aurora B and Survivin, whereas TRABID depletion in mitotic cells led to opposite effects (Supplementary Fig. 3c–f). TRABID knockdown also suppressed K29-polyubiquitination of endogenous Aurora B and Survivin in mitotic cells (Fig. 3e, f). To determine whether Aurora B and Survivin are direct substrates of TRABID, we used a baculovirally purified TRABID from a commercial source. The purified TRABID readily interacted with purified Aurora B and Survivin in vitro (Fig. 3g). Furthermore, in vitro deubiquitination assay demonstrated that purified TRABID diminished the K29-ubiquitination levels of Aurora B and Survivin (Fig. 3h, i). These data support that Aurora B and Survivin are direct substrates of TRABID.

To determine the consequence of TRABID-mediated deubiquitination of Aurora B and Survivin, we performed cycloheximide chase assay. Remarkably, TRABID knockdown in mitotically arrested cells potentiated the turnover of Aurora B and Survivin (Fig. 3j). Furthermore, proteasome inhibitor MG132 reversed TRABID depletion-mediated downregulation of Aurora B and Survivin in mitotically arrested cells (Fig. 3k). Thus, TRABID prevents Aurora B and Survivin from proteasomal degradation in mitosis. Notably, although TRABID did not affect the ubiquitination of INCENP and Borealin, TRABID knockdown or inhibition was capable of downregulating these two proteins. We examined whether Aurora B or Survivin degradation led to the destabilization of INCENP and Borealin. Indeed, Aurora B or Survivin knockdown each resulted in the downregulation of INCENP and Borealin (Supplementary Fig. 3g). Thus, our study identifies Aurora B and Survivin as TRABID substrates and TRABID-mediated

deubiquitination of Aurora B and Survivin leads to the stabilization of all CPC components through a direct or indirect mechanism.

## TRABID ablation induces micronuclei by mitosis and autophagy defects

Having demonstrated the critical role of TRABID in stabilizing CPC to ensure proper mitotic cell division, we next investigated the effect of TRABID depletion or inhibition on chromosome stability. Notably, besides Trabid KO MEFs, the micronuclei phenotype was also observed from HeLa cells expressing TRABID shRNAs or siRNA (Fig. 4a and Supplementary Fig. 4a). Re-expressing TRABID wild types, but not TRABIDΔN/CS, in TRABID knockdown cells reverted the micronuclei phenotype (Supplementary Fig. 4a). Furthermore, an increase of micronuclei was also observed in TRABID-depleted B16F10 cells and murine colon carcinoma CT26 cells as well as TRABID inhibitor-treated HeLa and B16F10 cells (Fig. 4b and Supplementary Fig. 4b–d), demonstrating a general effect of TRABID deficiency on the induction of cytosolic micronuclei. Of note, centromeres were detected in the micronuclei derived from TRABID-depleted HeLa cells (Fig. 4c), indicating the involvement of an aneugenic mechanism for the micronuclei formation[47,48]. However, since micronuclei can be removed by a selective autophagy process called micronucleophagy[49–52], autophagy defects may also contribute to a net increase of micronuclei. To evaluate the potential role of mitosis and autophagy defects in the micronuclei phenotype induced by TRABID ablation, we rescued TRABID-depleted cells with the corresponding substrates. Importantly, re-expression of Aurora B/Survivin or VPS34 in TRABID-knockdown HeLa cells each suppressed the micronuclei phenotype, whereas combined expression of all three proteins led to a further reduction of cells carrying micronuclei (Fig. 4d). To further substantiate the contribution of autophagy deficiency to micronuclei accumulation in TRABID-knockdown cells, we activated autophagy using rapamycin and found that this manipulation alleviated micronuclei phenotype (Supplementary Fig. 4e). Conversely, blockage of autophagic flux by bafilomycin A1 reverted the suppressive effect of VPS34 re-expression on micronuclei accumulation in TRABID-knockdown cells (Fig. 4e). In addition, the colocalization or encompassing of micronuclei with LC3, an indication of the micronucleophagy event, was greatly decreased by TRABID knockdown, which was partially rescued by VPS34 re-expression (Fig. 4f). Our data support that the deficiencies in mitosis and autophagy both contribute to the micronuclei phenotype seen in TRABID-depleted cells, likely through the increase of micronuclei generation and the decrease of their clearance, respectively. Data or images are representative of three (for b, c, d, e, g, h) or two (for f, i) independent experiments.

## TRABID deficiency prevents cGAS from autophagic degradation

The cytosolic micronuclei are sensed by cGAS, leading to the activation of cGAS/STING pathway[12,22,23,53]. Besides this well-known function of cGAS, a recent study identifies cGAS as a micronucleophagy receptor[52].

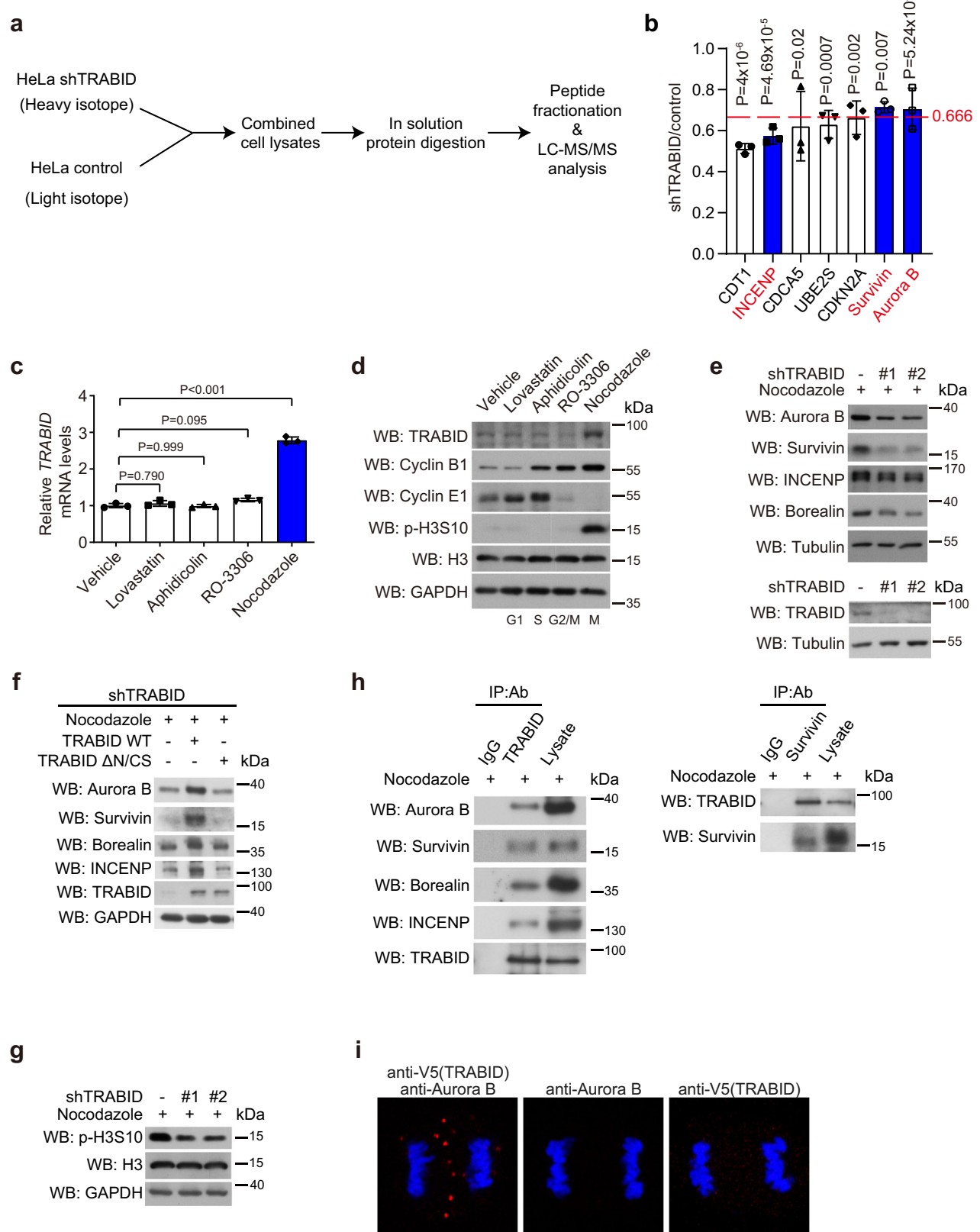

Interestingly, the cGAS-mediated micronucleophagy not only reduces micronuclei but promotes autophagic degradation of cGAS, thereby dampening cGAS-mediated innate immunity. These findings point out the dual roles cGAS in micronuclei-driven innate immunity. Since cGAS-mediated micronucleophagy requires the function of VPS34[52], we reasoned that TRABID inhibition should impair micronucleophagy

to prevent cGAS from autophagic degradation through VPS34 desta-bilization. Consistent with this idea, while Aurora B depletion reduced cGAS abundance through a lysosome-dependent mechanism (Fig. 5a), indicative of autophagic degradation, this phenomenon was not seen in TRABID-knockdown cells (Fig. 5b). Furthermore, re-expression of VPS34 in TRABID-knockdown cells restored cGAS degradation. Of

**Fig. 2 | TRABID is upregulated in M phase to control CPC abundance.**
**a** Experimental workflow of SILAC-based analysis of proteome changes by TRABID knockdown. **b** Mitotic cell division regulators that are significantly downregulated by TRABID depletion as revealed by LC-MS/MS. Data are mean ± SD, $n = 3$ independent experiments. *P* values are determined by two-sided Student's t-test.
**c, d** RT-qPCR (**c**) and Western blot (**d**) analyzes of *TRABID* mRNA and protein levels in HeLa cells treated with 10 μM lovastatin, 2 μg/ml aphidicolin, 10 μM RO-3306, or 3 μM nocodazole for 18 h. Data are mean ± SD, $n = 3$ independent experiments. *P* values are determined by one-way ANOVA with Tukey's post hoc test.
**e–g** Western blot analysis of indicated proteins in HeLa cells stably expressing

control or TRABID shRNAs together with or without TRABID wild type or TRABIDΔN/CS and synchronized in M phase by treatment with 3 μM nocodazole for 18 h. The knockdown efficiencies of shRNAs are shown in (**e**). **h** Reciprocal immunoprecipitation analyzes of the interactions between TRABID and CPC components in HeLa cells treated with 3 μM nocodazole for 18 h. **i** HeLa cells transiently expressing V5-TRABID were synchronized at G2 by treatment with 10 μM RO-3306 for 16 h and then released to proceed the cell cycle for 40 min before monitoring the PLA signal with indicated antibodies. Bar, 10 μm. Blots or images are representatives of three (for **d, e, g, h**) or two (for **f, i**) independent experiments. Source data are provided as a Source Data file.

note, the difference in cGAS degradation seen in Aurora B- and TRABID-depleted cells could not be attributed to an enhanced micronuclei phenotype in the former situation, since Aurora B knockdown displayed a slightly lower efficacy in micronuclei induction than TRABID knockdown (comparing Supplementary Fig. 5a with Fig. 4a). Furthermore, a similar lack of cGAS degradation was observed from TRABID inhibitor-treated cells, which was again recovered by VPS34 overexpression (Fig. 5c). To directly evaluate the effect of TRABID deficiency on cGAS autophagic degradation, we analyzed the lysosome accumulation of cGAS by lyso-IP[54], a previously established method for rapid isolation of lysosomes. The successful isolation of lysosomes without contaminating other intracellular organelles was demonstrated by Western blot with lysosome markers LAMP1 and LAMP2, Golgi marker GM130, mitochondria marker ATP5A or VDAC, and endosome marker EEA1. Remarkably, while Aurora B knockdown potentiated lysosome accumulation of cGAS, TRABID knockdown did not show any effect (Fig. 5d). Re-expression of VPS34 in TRABID-depleted cells stimulated cGAS accumulation in lysosomes (Supplementary Fig. 5b). Thus, our study describes a role of TRABID deficiency in preventing cGAS from autophagic degradation.

### TRABID deficiency activates cGAS/STING pathway
Since TRABID deficiency impairs autophagic degradation of micronuclei and cGAS, we reasoned that TRABID deficiency would switch the dual roles of cGAS in micronuclei-driven innate immunity to purely activate the cGAS/STING pathway. Accordingly, TRABID depletion or inhibition in HeLa cells increased cellular cGAMP level (Supplementary Fig. 5c, d), a readout of cGAS activity. Consequently, the levels of p-TBK1, p-IRF3, p-STING, and p-STAT1 were all elevated in TRABID-depleted HeLa cells (Fig. 5e, f and Supplementary Fig. 5e), indicating the simulation of cGAS/STING axis. Furthermore, TRABID knockdown in B16F10 cells similarly upregulated p-TBK1 and p-IRF3 (Supplementary Fig. 5f). Consistent with the cGAS/STING-dependent NFκB activation, TRABID depletion in HeLa cells increased IKK and p65 phosphorylation, IκB degradation, and p65 nuclear translocation (Fig. 5g, h).

We further found that TRABID depletion in HeLa cells induced the expression of a panel of type I interferon genes and interferon-stimulatory genes (ISGs), and elevated the secreted IFNβ and the IFNβ reporter activity (Fig. 5i–k), thus demonstrating the activation of type I interferon responses. A similar induction of the expression of type I interferon genes and ISGs was observed from Trabid-depleted B16F10 and CT26 cells as well as TRABID inhibitor-treated HeLa and B16F10 cells (Fig. 5l and Supplementary Fig. 5g–i), demonstrating a general effect of TRABID deficiency on the activation of type I interferon responses. Furthermore, TRABID knockdown in HeLa, B16F10, and CT26 cells, as well as TRABID inactivation in HeLa cells, elevated the expression of *PD-L1* mRNA (Fig. 5k, l and Supplementary Fig. 5g, h), consistent with the finding that *PD-L1* is a transcription target of NFκB[55]. TRABID knockdown in HeLa and B16F10 cells also stimulated the expression of PD-L1 protein (Fig. 5m and Supplementary Fig. 5j). Importantly, disruption of cGAS/STING pathway by cGAS inhibitor G140 or STING inhibitor C176 significantly suppressed type I interferon

responses induced by TRABID knockdown (Supplementary Fig. 5k, l), indicating that the activation of cGAS/STING pathway is the primary mechanism for TRABID deficiency-induced type I interferon responses. Finally, re-expression of Aurora B/Survivin or VPS34 each attenuated TRABID inactivation-induced type I interferon responses, whereas re-expression of all three proteins almost completely abrogated these responses (Fig. 5n). Together, these findings identify the function of TRABID deficiency in the activation of cGAS/STING pathway and both mitosis and autophagy substrates of TRABID, i.e., Aurora B/Survivin and VPS34, respectively, are involved in this function.

### TRABID deficiency shapes an anti-tumor immune microenvironment
Next, we determined whether TRABID deficiency-induced activation of cGAS/STING pathway could elicit anti-tumor responses by shaping tumor immune microenvironment. To this end, we used the poorly immunogenic B16F10 model by orthotopic transplantation of control or Trabid-knockdown B16F10 cells to syngeneic C57BL/6 mice. Trabid depletion significantly impeded tumor growth, reduced tumor weight, and prolonged mice survival (Fig. 6a–c and Supplementary Fig. 6a). To determine the role of Trabid catalytic and ubiquitin chain-binding functions in tumor growth, we reconstituted Trabid knockdown cells with Trabid wild type or TrabidΔN/CS and found a higher promoting effect of Trabid wild type on tumor growth, comparing with TrabidΔN/CS (Fig. 6d, e and Supplementary Fig. 6b, c). Immunohistochemistry (IHC) analysis of the tumor sections revealed that tumors derived from Trabid-depleted cells exhibited reduced proliferation and increased apoptosis, as evident by Ki67 and active caspase-3 staining, respectively (Supplementary Fig. 6d). Interestingly, despite harboring defects in mitosis, Trabid-depleted melanoma cells did not show significant reductions in proliferation in vitro (Supplementary Fig. 6e). Furthermore, when inoculating the same set of cells to immune-deficient Nude mice, Trabid knockdown only slightly decreased tumor growth and the effect was much weaker than what was found in C57BL/6 mice (comparing Fig. 6f, g and Supplementary Fig. 6f with Fig. 6a, b and Supplementary Fig. 6a). These findings suggest the involvement of tumor immune microenvironment in the tumor-suppressive effect of Trabid deficiency seen in C57BL/6 mice. We thus determined the impact of tumor-intrinsic Trabid depletion on tumor immune microenvironment by analyzing tumor-infiltrating immune cells using multicolor flow cytometry (see Supplementary Fig. 6g–i for gating strategies). We found a significant increase of CD45+ cells (total leukocytes) in tumors formed by Trabid-deficient B16F10 cells, compared with control tumors (Fig. 6h). Furthermore, Trabid depletion in tumors increased the frequencies of tumor-infiltrating CD4+ T cells and CD8+ T cells but decreased that of Treg cells (Fig. 6i–k). Among the CD8+ T cells, the population expressing granzyme B, representing the activated cytotoxic T cells, was increased in tumors formed by Trabid-deficient cells (Fig. 6l). We also observed that Trabid deficiency in tumors increased the frequency of tumor-infiltrating NK cells and the proportion of macrophages expressing an M1 marker MHCII but decreased those expressing an M2 marker CD206 (Fig. 6m–o). Thus, Trabid deficiency in tumors suppresses tumor growth and induces an anti-tumor

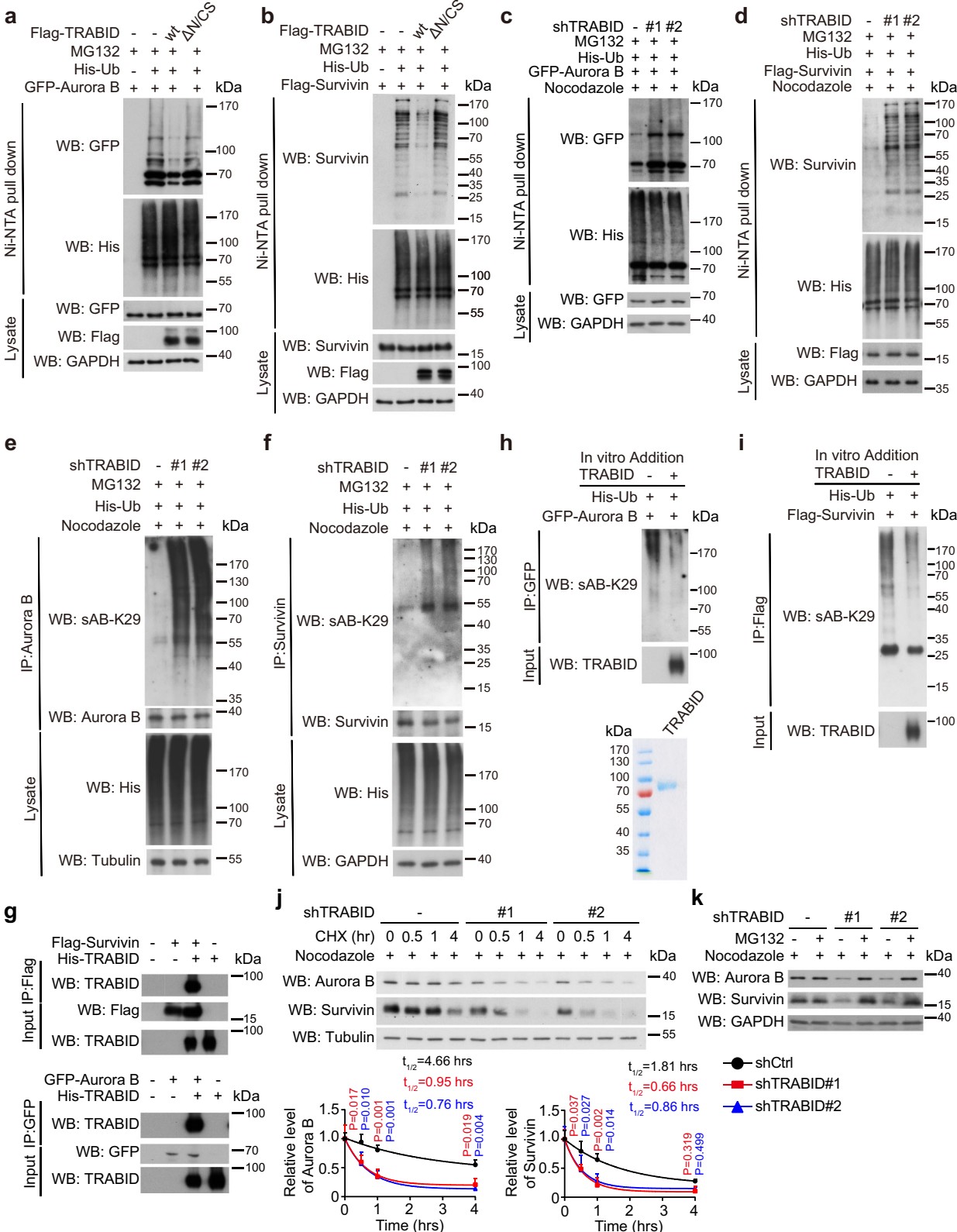

immune microenvironment by increasing the intratumoral infiltration of anti-tumor immune cells and decreasing that of pro-tumor immune cells.

To investigate whether the activation of cGAS/STING pathway participates in the anti-tumor effect of Trabid deficiency, we treated mice inoculated with Trabid-deficient B16F10 cells with a STING inhibitor C176. Remarkably, C176 significantly increased tumor growth and

tumor weight (Fig. 6p, q and Supplementary Fig. 6j), suggesting that stimulation of cGAS/STING axis contributes at least in part to the tumor-suppressive effect of Trabid deficiency.

**Trabid targeting sensitizes tumor response to anti-PD-1 therapy**
The enhancement of intratumoral infiltration of anti-tumor immune cells by Trabid depletion suggests that Trabid targeting could shape a

**Fig. 3 | TRABID deubiquitinates and stabilizes Aurora B and Survivin.**
**a**–**d** Western blot analysis of GFP-Aurora B and Flag-Survivin ubiquitination in 293 T cells transfected with indicated constructs (**a**, **b**) or HeLa derivatives as in Fig. 2e transfected with indicated constructs and treated with 3 μM nocodazole for 18 h (**c**, **d**). Cells were treated with MG132 to preserve the ubiquitination signals. **e**, **f** Analysis of the K29-linked ubiquitination on endogenous Aurora B and Survivin in HeLa derivatives as in Fig. 2e transfected with His-ubiquitin and treated with 3 μM nocodazole for 18 h. Cells were treated with MG132 to preserve the ubiquitination signals. **g** In vitro binding assay by incubating baculovirally purified His-TRABID with Flag-Survivin or GFP-Aurora B bound on beads. **h**, **i** In vitro deubiquitination assay. Aurora B or Survivin was immunoprecipitated from 293 T cells transfected with His-ubiquitin and GFP-Aurora B or Flag-Survivin and incubated with purified

His-TRABID. Aurora B and Survivin K29-ubiquitination was determined by Western blot with sAB-K29. The purity of His-TRABID was shown in (**h**). **j** Western blot analysis of Aurora B and Survivin in HeLa derivatives as shown in Fig. 2e, treated with 3 μM nocodazole for 18 h, and with 100 μg/ml cycloheximide for indicated time points. The levels of Aurora B and Survivin were normalized with that of internal control and plotted. Data are mean ± SD, n = 3 independent experiments. P values are determined by two-way ANOVA with Tukey's post hoc test. **k** Western blot analysis of Aurora B and Survivin in indicated HeLa derivatives treated with 3 μM nocodazole for 18 h and then with or without 5 μM MG132 for 6 h. For (**a**–**k**), blots are representatives of three independent experiments. Source data are provided as a Source Data file.

"hot" TME to improve tumor response to ICB therapy. Furthermore, the upregulation of PD-L1 by TRABID deficiency also predicts a similar effect of TRABID targeting. For targeting TRABID, we chose a clinically applicable strategy by utilizing its inhibitor. Of note, TRABID inhibitor almost completely reversed the effect of TRABID on Survivin deubiquitination but could not inhibit the capability of A20, another OTU-family DUB, to deubiquitinate ATG9A (Supplementary Fig. 7a), a known substrate of A20[56]. In addition, TRABID inhibitor could not further augment the micronuclei phenotype seen in Trabid KO MEFs (Supplementary Fig. 7b). These findings suggest the specificity and an on-target effect of TRABID inhibitor. Next, we treated B16F10 tumor-bearing mice with anti-PD-1 antibody, Trabid inhibitor, or the combination of these two agents (Fig. 7a). While treatment of Trabid inhibitor or anti-PD-1 antibody alone decreased tumor growth and tumor weight, and prolonged mice survival, the combined treatment offered significantly improved therapeutic effects compared with single treatment (Fig. 7b–d and Supplementary Fig. 7c). Western blot analysis substantiated that Trabid inhibitor reduced Aurora B and Survivin expression in B16F10 tumors, leading to a decreased histone H3S10 phosphorylation (Supplementary Fig. 7d). Trabid inhibitor treatment also upregulated the phosphorylation of Tbk1 and Irf3, indicating the activation of cGAS/STING axis. We also found that Trabid inhibitor or anti-PD-1 antibody each compromised the proliferation of tumor cells and increased their apoptosis, whereas combined treatment showed more potent effects (Supplementary Fig. 7e). Additionally, flow cytometry analysis revealed that each treatment increased intratumoral infiltration of CD45[+] cells and this effect was further enhanced by combined treatment (Fig. 7e). Furthermore, the proportions of tumor-infiltrating anti-tumor immune cells, such as CD4[+] T cells, CD8[+] T cells, activated CD8[+] T cells, NK cells, and M1 macrophages, were all greatly enhanced, whereas the proportions of anti-tumor immune cells, such as Treg and M2 macrophages, were markedly reduced by combined treatment (Fig. 7f–l). In most cases, administration of Trabid inhibitor or anti-PD-1 antibody alone displayed weaker effects than combined treatment. Thus, our study demonstrates that Trabid inhibitor enhances the anti-tumor effect of PD-1 antibody by shaping an inflamed TME.

### Clinical association of TRABID expression with anti-tumor immunity

To assess the clinical relevance of our findings, we explored the correlation between TRABID expression and interferon signaling in human cancers. Using a previously established method for calculating the ISG score derived from the expression profile of a 38-gene signature[57], we found that TRABID high expression in most TCGA cancer types correlated with a lower ISG score (Supplementary Fig. 8a–q). Furthermore, the expression of IFNB1 and the three Th1-type chemokines CXCL9, CXCL10, and CXCL11 correlated negatively with TRABID expression in many TCGA solid cancer types (Fig. 8a). These findings revealed a negative correlation of TRABID expression with interferon signaling in human cancers. Since Th1-type chemokines are crucial for immune cell trafficking to TME, we further

assessed the correlation between TRABID expression and intratumoral infiltration of various immune cells by querying the TIMER2.0 web server (http://timer.comp-genomics.org/). Remarkably, TRABID expression displayed negative correlations with the infiltration of various types of anti-tumor immune cells and positive correlations with pro-tumor immune cells across different solid cancer types (Fig. 8b, c). These data identify a robust negative association of TRABID expression with an anti-tumor immune microenvironment.

### Discussion

In this study, we identify a role of tumor-intrinsic TRABID in regulating TME by showing that TRABID deficiency induces micronuclei to activate cGAS/STING pathway, thereby shaping an anti-tumor immune microenvironment (Fig. 8d). The clinical relevance of our findings is further strengthened by the identification of inverse correlations of TRABID expression with an interferon signature and the infiltration of many anti-tumor immune cells across most solid cancer types. Consistent with these findings, targeting TRABID by a pharmacological inhibitor sensitizes tumors to anti-PD-1 therapy. Our study thus describes TRABID as a potential target for improving the efficacy of anti-cancer immunotherapy.

Intriguingly, the suppressive effect of TRABID on the activation of cGAS/STING pathway is mediated by a combinatory defect in mitosis and autophagy (Fig. 8d). TRABID deficiency compromises mitosis by enhancing the ubiquitination and proteasomal degradation of Aurora B and Survivin, resulting in the further downregulation of other CPC subunits. One consequence of such mitotic defect is CIN-mediated formation of cytosolic micronuclei. However, these micronuclei can be cleared by a selective autophagy process called micronucleophagy, in which cGAS acts as an autophagy receptor to link micronuclei to the LC3-containing autophagosomes[52]. This process leads to the autophagic degradation of both micronuclei and cGAS, thus serving as a checkpoint of innate immunity. Notably, VPS34 complex is required for micronucleophagy[52] and the complex component Beclin-1 binds cGAS to facilitate this selective autophagy process[58]. However, TRABID deficiency impairs autophagy by destabilizing the entire VPS34 complex[37], which would block micronucleophagy to relief this innate immunity checkpoint. Consistent with this notion, our study supports that the autophagy defect caused by TRABID ablation contributes in part to the micronuclei induction and prevents cGAS from autophagic degradation. Thus, the dual roles of TRABID in mitosis and autophagy could render it as a more potent target for activating cGAS/STING axis than those governing chromosome stability alone. Of note, even the effects of TRABID deficiency on the induction of micronuclei and interferon responses are reverted by combinatory expression of TRABID substrates Aurora B, Survivin, and VPS34, we do not exclude the possibility that other substrates of TRABID could also participate in the innate immune response induced by TRABID deficiency.

Given the promising role of cGAS/STING pathway in anti-cancer immunotherapy, many natural and synthetic STING agonists have been developed and tested in preclinical models and clinical trials[9–11]. Nevertheless, there are potential obstacles for the application of STING

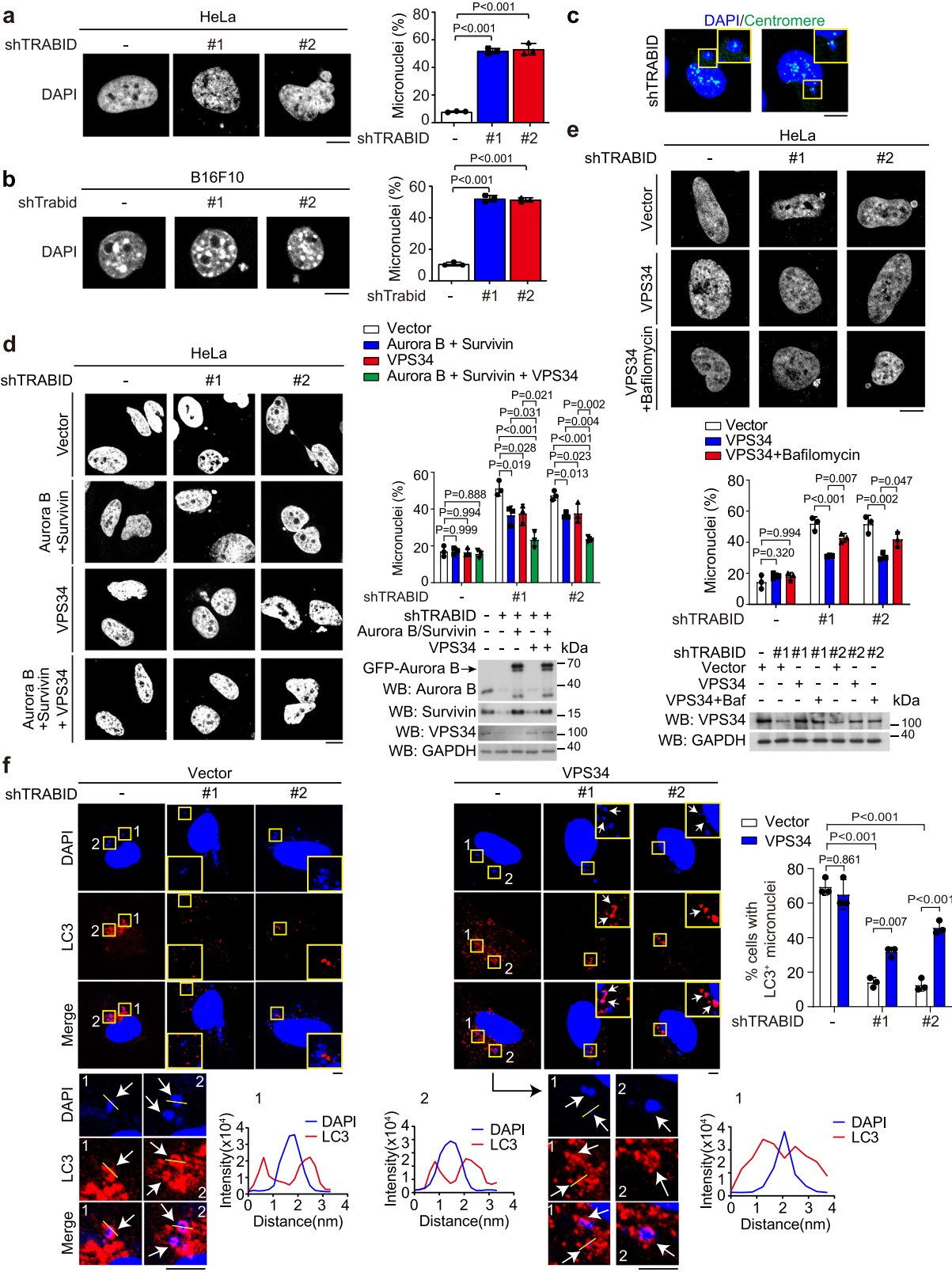

agonists to cancer therapy. One hurdle is the existence of several pro-tumor effects of this pathway. For instance, STING activation promotes metastasis through the non-canonical NFκB pathway[12], supports tumor cell survival through IL-6 pathway[14], and stimulates the proliferation of low-antigenicity tumors through the induction of tolerogenic responses[15]. Thus, selective inhibition of these pro-tumor functions upon the administration of STING agonists would be beneficial.

However, since TRABID elicits certain pro-tumor functions such as proliferation and metastasis[38–40], it might be an ideal target for not only inducing STING's anti-tumor effects but neutralizing its pro-tumor effects on metastasis and proliferation, thereby achieving a better therapeutic outcome than STING agonists. Furthermore, TRABID inhibition could compromise autophagy, which is often exploited by tumor cells for maintaining their growth and survival in the harsh

**Fig. 4 | TRABID deficiency induces micronuclei through mitosis and autophagy defects. a, b** HeLa or B16F10 cells stably expressing control or TRABID shRNAs were stained with DAPI. Representative images and the percentages of cells with micronuclei are shown. Bar, 10 μm. **c** FISH analysis of centromeres on HeLa cells stably expressing TRABID shRNA. Boxed areas are enlarged to show micronuclei with centromere signals. Bar, 10 μm. **d, e** HeLa cells stably expressing control or TRABID shRNAs were transiently transfected with indicated constructs and treated with or without 100 nM bafilomycin A1 for 16 h. Cells were stained with DAPI and then examined for micronuclei by confocal microscopy. Representative images, the percentages of cells with micronuclei, and the expression levels of various proteins are shown. Bar, 10 μm. Blots are representatives of two independent experiments. Notably, GFP-Aurora B was used for transfection in d and its position is marked. **f** HeLa cells stably expressing control or TRABID shRNAs were transiently transfected with VPS34 and then stained with LC3 antibody and DAPI. Representative images are shown. Boxed areas are enlarged and shown in the inset or below. Bars, 5 μm. Micronuclei that are colocalized or surrounded by LC3 signals are indicated by arrows. Yellow lines indicate the paths along which the relative intensities of LC3 and DAPI were quantified and plotted (labeled with "1" or "2" to indicate the corresponding images). The percentages of cells with LC3⁺ micronuclei are shown. Data in (**a**), (**b**), (**d**), (**e**), (**f**) are mean ± SD (*n* = 3 independent experiments and >20 cells per group per experiment were counted). *P* values are determined by one-way (**a**, **b**, **d**, **e**) or two-way (**f**) ANOVA with Tukey's post hoc test. Source data are provided as a Source Data file.

TME[59,60]. Thus, TRABID inhibition likely confers multifaceted tumor-suppressive effects.

Of note, a previous study indicated that germ-line deletion of *Zranb1* shows no or minor effects on the development of various cells in the immune system[61], suggesting that Trabid may not be an indispensable factor for the mitosis of immune cells. Perhaps other DUBs expressed in these cells could compensate Trabid loss for stabilizing CPC. This finding might also imply the devoid of major adverse effects by TRABID inhibition. Nevertheless, the same study found that *Trabid* KO in dendritic cells (DCs), but not T cells, attenuates autoimmune inflammation in an T cell-dependent experimental autoimmune encephalomyelitis model, even though Trabid KO does not affect the development, migration and maturation of DCs[61]. Mechanistically, Trabid deficiency impairs the production of inflammatory cytokines IL-12 and IL-23 by DCs through epigenetic regulation. Thus, TRABID likely plays distinct roles in immune regulation under different disease contexts, such as in the autoimmune system and TME, by targeting different substrates. Future studies are needed to further dissect the physiological and pathological functions of TRABID. Furthermore, it would be important to further characterize the effects of TRABID inhibitors, such as their specificity, safety, and tolerability.

We identify CPC components Aurora B and Survivin as the substrates of TRABID and TRABID-mediated deubiquitination prevents the two proteins from proteasomal degradation. Of note, other DUBs, such as USP35[33] and USP13[32], have been reported to stabilize Aurora B through deubiquitination. However, evidence has emerged that USP35, USP13, and TRABID do not merely act on Aurora B in a completely redundant fashion. For instance, USP13 mainly stabilizes Aurora B in the interphase, prior to the onset of mitosis. TRABID and USP35, however, should mainly stabilize Aurora B in mitotic cells as the former is upregulated in mitosis and the latter is induced by FoxM1, a transcription factor controlling the expression of many G2/M genes[62]. Nevertheless, USP35 could antagonize APC^Cdh1-mediated ubiquitination on Aurora B[33]. However, TRABID specifically disassembles K29- and K33-linked ubiquitin chains[36] as well as K29/K48 branched ubiquitin chains[37] and therefore would be unfavorable to reverse the action of APC^Cdh1, which assembles K11-containing branched ubiquitin chains[63,64]. Thus, USP35 and TRABID may stabilize different or at least non-overlapping pools of Aurora B. Furthermore, since APC^Cdh1 governs Aurora B degradation as cells exit mitosis, DUB should act on Aurora B prior to the action of APC^Cdh1 in order to stabilize Aurora B in mitosis, thereby ensuring a proper mitotic procession. Thus, from chain type-specificity and action-timing points of view, TRABID likely counteracts the function of an Aurora B E3 ligase other than APC^Cdh1. Future studies will aim to identify this E3 ligase. Finally, given the critical role of USP35 and TRABID in mitosis by stabilizing Aurora B, we believe that protein stabilization represents an important mechanism for Aurora B induction in mitosis, apart from the well-known transcription regulatory mechanism.

We show that TRABID also acts on Survivin, and this co-regulation of Aurora B and Survivin resembles the function of USP35[33,35]. Of note, Survivin is present in the cytoplasm of interphase cells, where it elicits potent and multifaceted oncogenic effects such as inhibition of apoptosis and promotion of migration, angiogenesis, and stemness[65]. Consistent with these pro-tumor functions, Survivin is overexpressed in many cancer types[66]. It is unclear whether TRABID stabilizes Survivin in the interphase to enhance these pro-tumor functions, which warrants further analysis.

Using the K29-ubiquitin chain-specific binder sAB-K29[46], we show that Aurora B and Survivin are modified by this atypical ubiquitin chain and that TRABID is capable of removing such modification from the two proteins. Interestingly, while our study proposes a role of TRABID-mediated K29-deubiquitination in the progress of late mitotic stages, a recent study reported a requirement of HECTD1, an E3 ligase specifically assembling K29/K48 branched ubiquitin chains[67], in the progression of cell cycle through early mitotic stages[68]. Furthermore, enrichment of K29-ubiquitination in midbody was reported, which is implicated in the midbody assembly during telophase[46]. Thus, the current and previous studies highlight the delicate controls of K29-ubiquitination and deubiquitination events for proper progression through the different mitotic stages.

Given the induction of both TRABID and Aurora B in M phase, TRABID-mediated Aurora B stabilization should mainly occur during mitosis. TRABID deficiency leads to mitotic errors, which facilitate the generation of micronuclei after nuclear envelop reforms at the end of mitosis. The subsequent cGAS/STING activation should occur only in the interphase, as cGAS is inactivated in M phase through hyperphosphorylation mediated by Aurora B[69]. Thus, TRABID deficiency-induced Aurora B destabilization might impede cGAS inactivation in M phase to contribute in part to the anti-tumor immunity. In addition, the micronucleophagy might also occur mainly in the interphase, as autophagy is suppressed in M phase through the inactivation/degradation of several key autophagic proteins[70,71]. Thus, the TRABID deficiency-induced anti-tumor immunity described in this study involves several steps occurring at different stages of cell cycle.

In sum, our study identifies a critical role of TRABID in mitotic cell division. This effect, in conjunction with its autophagy-promoting function, making TRABID a potent suppressor of cGAS/STING signaling. Targeting TRABID is a promising strategy for activating cGAS/STING-dependent anti-tumor immunity, thereby suppressing tumor growth and improving anti-cancer immunotherapy.

## Methods

### Plasmids

Plasmids encoding His-ubiquitin, V5-TRABID, V5-TRABID C443S, and V5-VPS34 were described previously[37]. cDNAs for wild type TRABID and TRABIDΔN/CS (Δ1-32; C443S) were cloned to pRK5F and pLAS5w. Plasmid encoding GFP-Aurora B was kindly provided by Tang K Tang (Academia Sinica, Taipei, Taiwan), whereas plasmids encoding pLuc-IFNβ, pTK-Renilla, V5-cGAS, and Myc-Flag-STING were gifts from Li-Chung Hsu (National Taiwan University, Taipei, Taiwan). pJM41-TMEM-192-Flag was a gift from Ping-Hung Chen (National Taiwan University, Taipei, Taiwan), whereas plasmids encoding Flag-TRAF6, EGFP-A20, V5-His-ATG9A, and Myc-ubiquitin were from Guang-Chao Chen

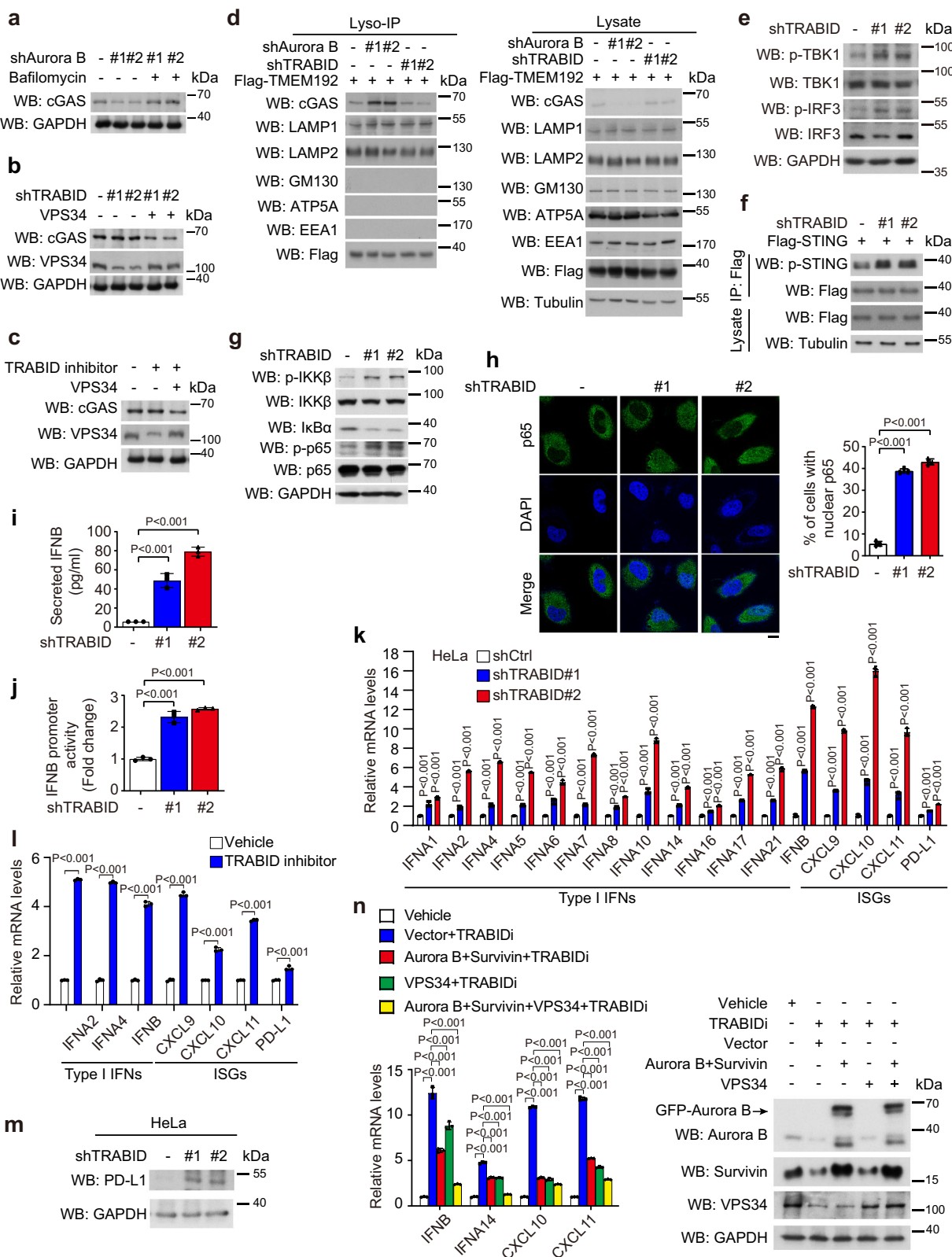

(Academia Sinica, Taipei, Taiwan). The plasmid encoding sAB-K29 is a kind gift from Minglei Zhao and Anthony Kossiakoff (University of Chicago, Chicago, IL). The cDNA fragments for Survivin, INCENP, and Borealin were amplified from plasmids pLENTI_TetOn_Hu-BIRC5 (Addgene #136348), mCherry-INCENP (Addgene #108487), and pJAG98-Borealin (Addgene #69741), respectively, and subcloned to pRK5-Flag. Human H2B cDNA was amplified from U2OS cDNA library

and cloned to the mCherry-C1. The H2B-mCherry was then subcloned to pBabe-puro3.

**Cell culture, MEF immortalization, and transfection**
HeLa (CCL-2) and 293 T (CRL-3216) cell lines were obtained from the American Type Culture Collection (ATCC, Manassas, VA, USA), whereas 293FT cells (R70007) were obtained from Thermo Fisher.

**Fig. 5 | TRABID deficiency attenuates cGAS autophagic degradation and activates cGAS/STING axis. a, b** Western blot analysis of cGAS levels in HeLa derivatives as in Supplementary Fig. 3g and treated with or without 100 nM Bafilomycin A1 for 16 h or TRABID shRNA-expressing HeLa cells as in Fig. 2e transiently transfected with VPS34. **c** Western blot analysis of cGAS levels in HeLa cells transfected with or without VPS34 and treated with 3 μM TRABID inhibitor for 24 h. **d** Lyso-IP analysis of cGAS levels in lysosomes isolated from HeLa cells stably expressing Aurora B shRNAs or TRABID shRNAs and transiently transfected with Flag-TMEM192. **e, g, m** Western blot analysis of indicated proteins in HeLa cells stably expressing control or TRABID shRNAs. **f** Flag-STING was immunoprecipitated from HeLa cells stably expressing control or TRABID shRNAs and transfected with Flag-STING, followed by Western blot analysis with indicated antibodies. **h** Immunofluorescence staining of p65 in HeLa cells stably expressing control or TRABID shRNAs. Representative confocal images and quantitative data are shown. Bar, 10 μm. Data are mean ± SD (n = 3 independent experiments and >30 cells per

group per experiment were counted). P values are determined by one-way ANOVA with Tuckey's post hoc test. **I, j** ELISA for IFNβ secretion (**i**) or luciferase assay for IFNβ promoter activity (**j**) of HeLa cells stably expressing indicated shRNAs. **k, l, n** RT-qPCR analysis of indicated genes in HeLa cells stably expressing control or TRABID shRNAs (**k**), HeLa cells treated with vehicle or 3 μM TRABID inhibitor for 24 h (**l**), or HeLa cells transfected with indicated constructs and treated with vehicle or 3 μM TRABID inhibitor for 24 h (**n**). Data are normalized to the control cells or untreated control and expressed as fold changes. The expression levels of various proteins are shown in (**n**). The position of GFP-Aurora B, which was used for transfection, is marked. Data in (**i**), (**j**), (**k**), (**l**), (**n**) are mean ± SD, n = 3 independent experiments. P values are determined by two-sided Student's t test (**l**), or one-way ANOVA with Tuckey's post hoc test (**i, j, k, n**). Blots are representatives of three (for **a–c** and **e–g**) or two (for **d, m, n**) independent experiments. Source data are provided as a Source Data file.

B16F10 (ATCC CRL-6475) and CT26 (ATCC CRL-2638) cells were provided by Che-Ming Jack Hu (Academia Sinica, Taipei, Taiwan), whereas Atg5 KO MEFs were obtained from Guang-Chao Chen (Academia Sinica, Taipei, Taiwan). Primary MEFs were isolated from E12.5 *Zranb1*<sup>flox/flox</sup> mouse (EM07669, The European Mouse Mutant Archive), plated onto 75 cm² culture flasks, and cultured to reach confluency. Next, $1 \times 10^6$ cells were seeded onto a 100-mm dish and passaged every 3 days until the cells were immortalized. HeLa cells were cultured in minimum essential medium supplemented with 10% fetal bovine serum (FBS) and 1% penicillin/streptomycin (P/S; Gibco, Thermo Fisher Scientific). 293 T and 293FT cells were maintained in Dulbecco's Modified Eagle's medium (DMEM) supplemented with 10% FBS and 1% P/S. B16F10 cells were cultured in DMEM high-glucose medium supplemented with 10% FBS and 1% P/S. CT26 cells were cultured in RPMI1640 medium supplemented with 0.1 M sodium pyruvate, 10% FBS, and 1% P/S. MEFs were maintained in DMEM supplemented with 10% FBS, 0.2 M L-glutamine (Thermo Fisher Scientific), 1× nonessential amino acids, 0.1 M sodium pyruvate, and 1% P/S. Transfection of HeLa cells was performed using Lipofectamine 2000 or 3000 reagent (Invitrogen), whereas transfection of 293 T and 293FT cells was conducted by the calcium phosphate method.

**Antibodies and reagents**
The antibodies against TRABID were described previously[37]. Other antibodies used in this study are listed in Supplementary Table 1. MG132 was purchased from Calbiochem. Rapamycin, Cycloheximide, Bafilomycin A1, and TRABID inhibitor NSC112200 were purchased from Sigma-Aldrich. Lovastatin, aphidicolin, RO-3306, nocodazole, cGAS inhibitor G140 and STING inhibitor C176 were obtained from MedChemExpress. Adenoviruses AdCre and AdLuc were obtained from Baylor College of Medicine Vector Develop Laboratory.

**RNA interference and lentiviral transduction**
Lentivirus-based shRNA constructs were purchased from RNA Technology Platforms and Gene Manipulation Core Facility (Taipei, Taiwan). The target sequences of various shRNAs are listed in Supplementary Table 2. Lentiviral transduction was described previously[37].

**Purification of sAB-K29**
The synthetic antigen-binding fragment (sAB-K29) was purified using a published protocol[46]. The sAB-K29-encoded plasmid was transformed into the *E. coli* RIL cells, and a single colony was selected and cultured in LB media with ampicillin. sAB-K29 expression was induced by adding 0.6 mM IPTG when the optical density at 600 nm reached 0.8 and continued for 16 h at 16 °C. The cell lysate was sonicated, and the supernatant was collected by centrifugation and loaded onto a prepacked protein G Sepharose 4 fast flow column (Cytiva). The column was washed with buffer A (20 mM Tris-HCl pH 7.4, 500 mM NaCl) and

eluted with buffer B (0.1 M glycine pH 2.7). The eluted sAB-K29 was immediately diluted by adding 4 times of its volume of buffer A and dialyzed overnight at 4 °C with buffer C (20 mM Tris-HCl pH 7.4, 150 mM NaCl). The dialyzed sAB-K29 was then purified using a Superdex 75 increase 10/300 GL column on an Akta Pure M FPLC at 4 °C (Cytiva). The pure fractions, confirmed by SDS-PAGE, were combined and concentrated to 1 mg/ml. A concentration of 1 μg/ml was used for Western blot.

**Western blot and immunoprecipitation**
Cells were lysed with RIPA lysis buffer containing 150 mM NaCl, 20 mM Tris-HCl (pH 7.5), 1% NP40, 0.1% SDS, 1% sodium deoxycholate, 1 μg/ml aprotinin, 1 μg/ml leupeptin, and 1 mM phenylmethylsulphonyl fluoride (PMSF). Western blot analysis with lysates containing an equal amount of proteins was described previously[37]. For the Western blot with sAB-K29 as the primary antibody, F(ab')2 fragment-specific antiserum (Jackson ImmunoResearch Laboratories) was used as the secondary antibody. For immunoprecipitation, cell lysates containing an equal amount of proteins were incubated with anti-Flag agarose beads (Sigma-Aldrich), GFP-Trap agarose beads (Chromotek), or antibody-conjugated Protein A Magnetic beads (Merck) at 4 °C for 2 h. The beads were washed with lysis buffer for three times and the bound proteins were analyzed by Western blot.

**Lyso-IP**
15 plates of cells transfected with Flag-TMEM192 were trypsinized, washed with phosphate-buffered saline (PBS), and centrifuged to obtain cell pellet. The pellet was resuspended with 1 ml homogenization buffer (Microsome Isolation Kit, BioVision), passed through a 30 G needle for 20 times, and then added the homogenization buffer to 9 ml. After centrifugation at $10,000 \times g$ for 15 min, the supernatant was centrifuged again at $20,000 \times g$ for 20 min to obtain the microsomal membrane pellet. The pellet was extracted with RIPA lysis buffer and proceeded immunoprecipitation.

**In vivo deubiquitination assay**
Cells transfected with His-ubiquitin and other constructs were treated with or without 3 μM nocodazole for 16 h followed by 5 μM MG132 (Calbiochem) for 6 h. The cells were lysed under denaturing conditions by buffer A containing 6 M guanidine-hydrochloride, 0.1 M Na$_2$HPO$_4$/NaH$_2$PO$_4$ (pH 8.0), and 10 mM imidazole. Lysates were incubated with Ni-NTA agarose (GE Healthcare) for 2 h at 4 °C. The beads were washed five times with buffer A/TI [1 vol buffer A: 3 vol buffer TI (25 mM Tris-HCl, pH 6.8 and 20 mM imidazole)] and five times with buffer TI, followed by Western blot analysis. Alternatively, cells were lysed with RIPA lysis buffer and the overexpressed or endogenous Aurora B or Survivin was isolated by immunoprecipitation, followed by Western blot with sAB-K29. In all experiments, the equal expression of His-ubiquitin was verified by Western blot.

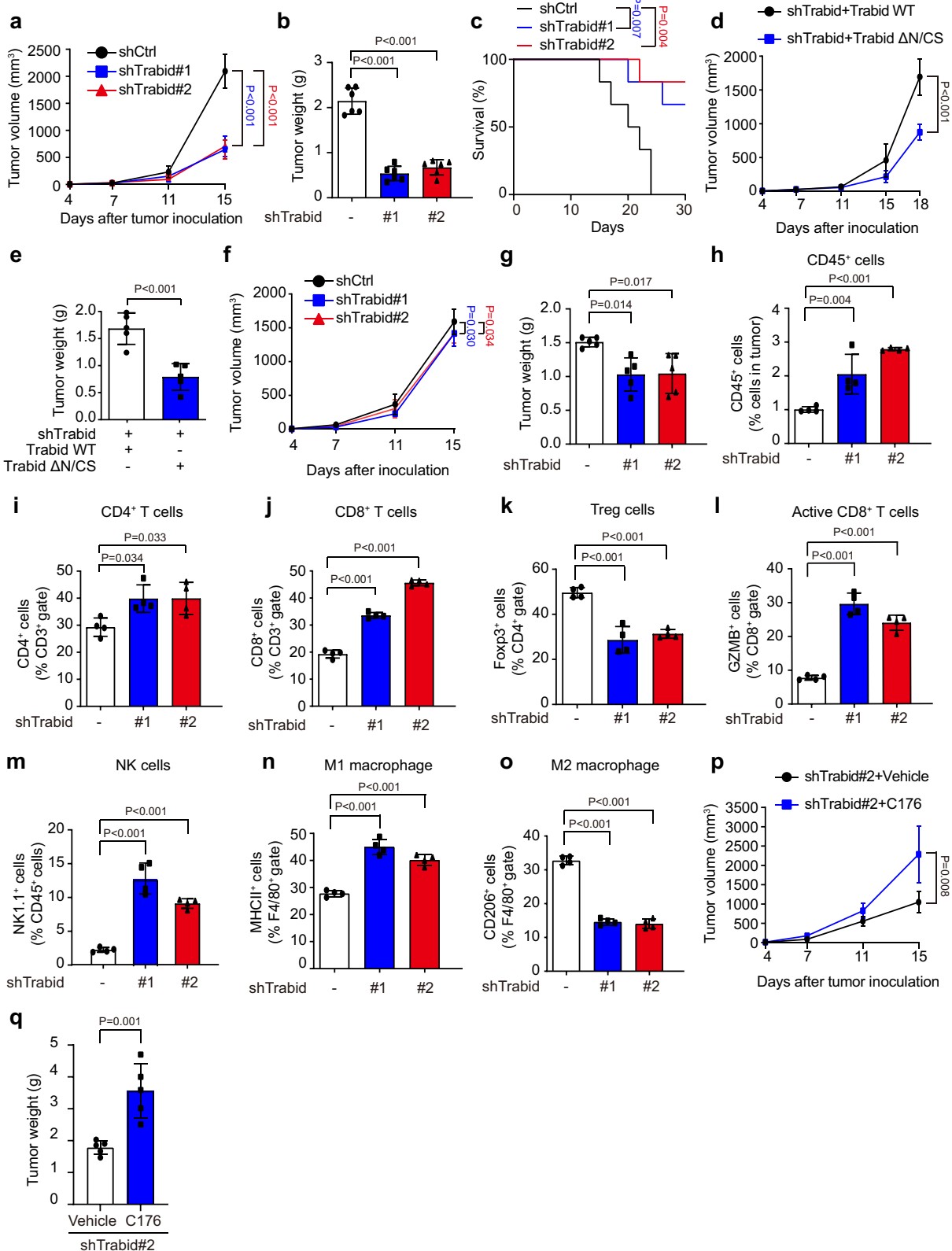

**In vitro deubiquitination assay**

A total of 293 T cells were transfected with His-ubiquitin and GFP-Aurora B or Flag-Survivin and ubiquitinated Aurora B or Survivin was immunoprecipitated from cell lysates with GFP-Trap agarose beads (Chromotek) or anti-Flag agarose beads (Sigma-Aldrich), respectively. Proteins bound on beads were incubated with 2 μM TRABID (Boston Biochem) at 37 °C for 1 h in 30 μl reaction mixture containing 20 mM Tris-HCl (pH 7.6), 150 mM NaCl, 0.1% Triton X-100, 0.2% NP-40, and 10 mM DTT. The beads were washed with RIPA lysis buffer for 5 times and then analyzed by Western blot with the sAB-K29 binder.

**PLA**

Cells were fixed with 4% formaldehyde for 15 min at room temperature, permeabilized with 0.1% Triton X-100 for 15 min at room temperature,

**Fig. 6 | TRABID knockdown in tumor cells induces an anti-tumor immune microenvironment. a, b** Tumor volume at indicated days and tumor weight at day 15 after C57BL/6 mice inoculated with B16F10 cells expressing control or Trabid shRNAs. Data are mean ± SD ($n = 6$ per group). $P$ values are determined by two-way (**a**) or one-way (**b**) ANOVA with Tukey's post hoc test. **c** Kaplan–Meier survival curves for C57BL/6 mice inoculated with B16F10 cells expressing control or Trabid shRNAs. P values are determined by log rank Mantel Cox test, $n = 6$. **d, e** Tumor volume at indicated days and tumor weight at day 18 after C57BL/6 mice inoculated with B16F10 cells expressing Trabid shRNAs and reconstituted with Trabid wild type or ΔN/CS mutant. Data are mean ± SD ($n = 5$ per group). $P$ values are determined by two-side Student's t-test. **f, g** Tumor volume at indicated days and tumor weight at day 15 after Nude mice inoculated with B16F10 cells expressing control or Trabid shRNAs. Data are mean ± SD ($n = 5$ per group). $P$ values are determined by two-way (**f**) or one-way (**g**) ANOVA with Tukey's post hoc test. **h−o** Flow cytometry analysis for the percentages of indicated types of tumor-infiltrating immune cells in the tumor tissues taken from day 15 of C57BL/6 mice after inoculation with B16F10 cells expressing control or Trabid shRNAs. Data are mean ± SD ($n = 4$ per group). $P$ values are determined by one-way ANOVA with Tukey's post hoc test. (**p, q**) Tumor volume at indicated days and tumor weight at day 15 after C57BL/6 mice inoculated with B16F10 cells expressing Trabid shRNA and treated with vehicle or C176. Data are mean ± SD ($n = 5$ per group). $P$ values are determined by two-sided Student's $t$ test. Source data are provided as a Source Data file.

and blocked with Duolink Blocking Solution for 1 h at room temperature. Cells were incubated with the mixture of two primary antibodies for 1.5 h at room temperature. After washing, Duolink secondary antibodies were added and incubated in a 37 °C humidified chamber for 1 h. Proximity ligation was performed with ligation for 30 min at 37 °C, followed by amplification for 1 h at 37 °C according to the manufacturer's protocol. The samples were then mounted for confocal analysis.

### Fluorescence in situ hybridization (FISH)
FISH was performed with the PNA FiSH_Human Pan-centromere FISH Probes (PNA Bio Inc.) according to the manufacturer's instructions. Briefly, cells were fixed by paraformaldehyde and permeabilized with Protease K buffer containing 10 mM Tris-HCl (pH 8.0), 0.1 M NaCl, 1 mM EDTA, 100 μg/ml RNase A, and 50 μg/ml protease K for 40 min. Then, the slides were dehydrated by consecutive incubations with 75%, 95%, and 100% ethanol and hybridized with 500 nM Alexa 488-labeled Human only pan-centromere probes (5′-AAACTAGACAGAAGCATT-3′) in hybridizing solution (60% formamide, 0.5% blocking reagent (Roche), 20 mM Tris-HCl pH 7.4) at room temperature for 2 h. After washing out the unlabeled probes, DNA was counterstained by 0.2 μg/ml DAPI at room temperature for 10 min.

### Immunofluorescence
Cells seeded on coverslips were washed three times with PBS and fixed with 4% formaldehyde at room temperature for 20 min. After three washes with PBS, cells were permeabilized with ice-cold methanol for 10 min and then washed three times with PBS. Cells were blocked in PBS containing 1% BSA and 10% goat serum at room temperature for 1 h and incubated with primary antibody diluted in blocking buffer at 4 °C overnight. Next, cells were washed three times with PBS, rinsed once with blocking buffer, and then incubated with fluorescent dye-conjugated secondary antibody (Life Technologies) together with DAPI (1 μg/ml) (Sigma-Aldrich) at room temperature for 30 min. Cells were washed three times with PBS and mounted with a mounting medium (Dako).

### Confocal microscopy and image assay
Fluorescence-stained cells were examined by a confocal microscope (Olympus FV3000) equipped with a 60x/1.40 oil objective lens (Olympus Objective Lens, PlanApo N) and images were collected by an OLYMPUS FV3000 FV31S-SW (v 2.40) software. To quantify the percentage of cells with micronuclei, nucleus images were defined and analyzed by Image J 1.53c, followed by counting manually.

### Time-lapse microscopy
Trabid KO MEFs or TRABID knockdown HeLa cells and their control counterparts were transduced with lentivirus-carrying H2B-mCherry. The stable cells were incubated in a micro-cultivation system with temperature and CO₂ controlled devices (Carl Zeiss). The cells were monitored on an inverted microscope (Axio Observer.Z1/7, Carl Zeiss) using an LD Plan-NEOFLUAR 20x NA 0.4 Korr Ph2 M27 objective lens.

Images were captured every 2 min for 48 h using a digital camera (ORCA, Hamamatsu) and were processed by the ZEISS ZEN2 image software (ZEN 2.6 blue edition).

### Reverse transcription-quantitative PCR (RT-qPCR)
Total RNAs were extracted using the Trizol reagent (Invitrogen). cDNAs were synthesized using the iScript cDNA Synthesis Kit (Bio-Rad, Hercules, CA, USA). Quantitative real-time PCR (qPCR) was performed using the SYBR Green PCR Master Kit (Applied Biosystems) and amplification was conducted on the Roche LightCycler 480 system. Gene expression levels were normalized to the level of housekeeping gene GAPDH. The sequences of PCR primers are listed in Supplementary Table 3.

### SILAC labeling
Cells were cultured for 2 weeks in SILAC medium containing DMEM deficient in L-Arginine and L-Lysine (Thermo Fisher Scientific) and supplemented with 10% dialyzed FBS (Biological Industries), 1% P/S, and L-Arginine-HCl [$^{13}C_6$, $^{15}N_4$] (Arg-10)/L-Lysine-HCl [$^{13}C_6$] (Lys-6), or L-Arginine (Arg-0)/L-Lysine (Lys-0) (Thermo Fisher Scientific). Next, $1 \times 10^8$ cells for each population were seeded for growing overnight and then washed twice with PBS before harvest.

### Cell lysis, protein digestion, and peptide fractionation
Cell pellets were lysed using urea lysis buffer containing 9 M urea, 50 mM Tris-HCl (pH 8.0), 150 mM NaCl, 1 mM EDTA, and protease inhibitor (Roche). Lysates were centrifuged to remove cell debris and lysates with an equal amount of proteins from two samples were mixed, reduced with 10 mM dithiothreitol (Sigma-Aldrich) at 37 °C for 1 h, and subsequently alkylated with 25 mM iodoacetamide (Sigma-Aldrich) at room temperature for 1 h in the dark. Lysates were diluted to 4 M urea with 50 mM Tris-HCl (pH 8.0), and proteins were digested with Lys-C (Wako) at 37 °C for 4 h. Next, the peptide mixtures were diluted to 1 M urea and subsequently digested with trypsin (Promega) for 16 h at 37 °C. The reaction was quenched with 0.1% formic acid and then cleared by centrifugation. Peptides were desalted using the Sep-Pak C18 Plus Cartridge (Waters) and an aliquot of 50 μg eluent was lyophilized for fractionation. Peptide fractionation was performed according to the manufacturer's instruction (Thermo). Briefly, dried peptides were reconstituted in 0.1% trifluoroacetic acid solution and then loaded onto a high-pH, reversed-phase fractionation spin column. Peptides were bound to the hydrophobic resin under aqueous conditions and desalted by washing the column with water. Subsequently, bound peptides were eluted into eight different fractions collected by centrifugation with a step gradient of increasing acetonitrile concentration.

### LC-MS/MS analysis
NanoLC-nanoESi-MS/MS analysis was performed on a Thermo UltiMate 3000 RSLCnano system connected to a Thermo Orbitrap Fusion mass spectrometer (Thermo Fisher Scientific, Bremen, Germany) equipped with a nanospray interface (New Objective, Woburn, MA)

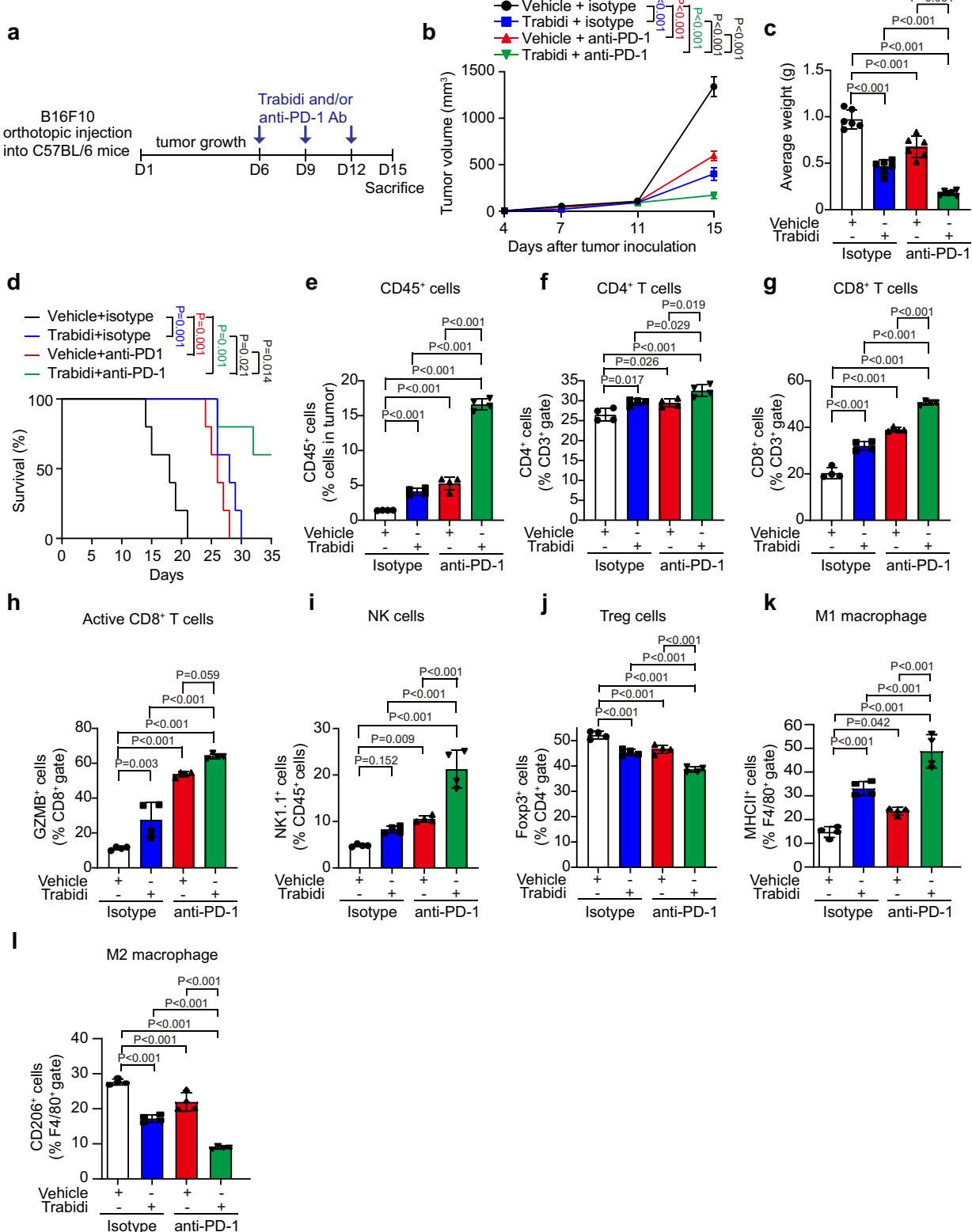

**Fig. 7 | TRABID targeting enhances the anti-tumor functions of PD-1 antibody.**
**a** Schematic presentation of the experimental workflow. C57BL/6 mice subcutaneously inoculated with B16F10 cells were injected with anti-PD-1 antibody and/or TRABID inhibitor as indicated. **b, c** Tumor volume at indicated days and tumor weight at day 15 are plotted. Data are mean ± SD ($n = 6$ animals). $P$ values are determined by two-way ANOVA (**b**) or one-way ANOVA (**c**) with Tukey's post hoc test. **d** Kaplan-Meier survival curves for C57BL/6 mice bearing B16F10 tumors and treated as in (**a**). $P$ values are determined by log rank Mantel Cox test, $n = 5$. **e–l** Flow cytometry analysis for the percentages of indicated types of tumor-infiltrating immune cells in the tumor tissues taken from day 15 of C57BL/6 mice after inoculation with B16F10 cells and treated as in (**a**). Data are mean ± SD ($n = 4$ per group). $P$ values are determined by one-way ANOVA with Tukey's post hoc test. Source data are provided as a Source Data file.

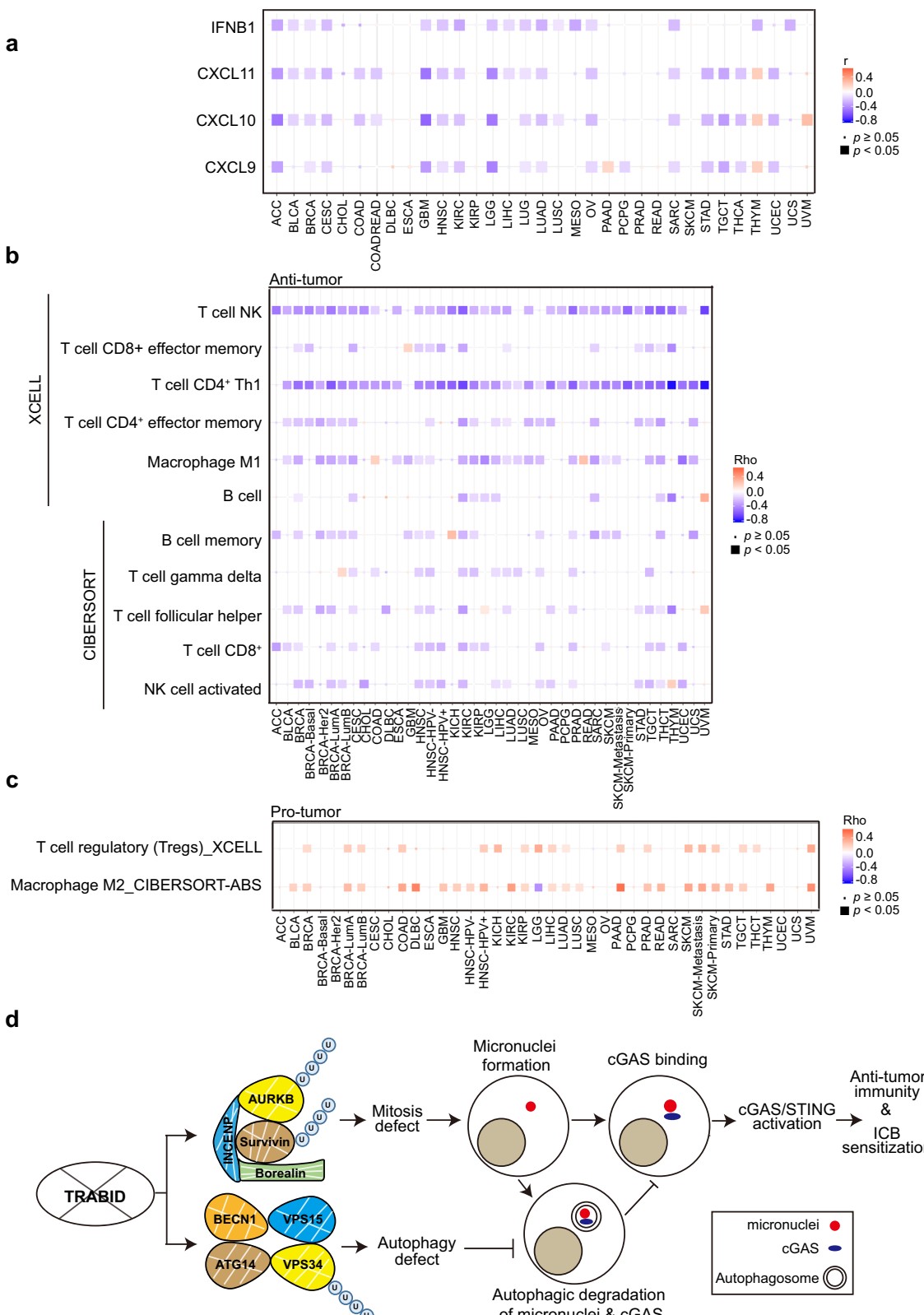

**Fig. 8 | TRABID expression correlates negatively with an anti-tumor immune microenvironment across diverse cancer types. a** Heatmap showing the correlation of TRABIID gene expression level with the expression levels of indicated genes. Data were retrieved from TCGA datasets. Cancer types are labeled on the x axis. *P* value are determined by Pearson's correlation. **b**, **c** Heatmap showing the correlation of TRABID gene expression levels with the infiltration of indicated anti-tumor (**b**) or pro-tumor (**c**) immune cells. Correlations were obtained through the TIMER2.0 website. Cancer types and immune cells types are labeled on the x and y axes, respectively. *P* value are determined by Spearman's correlation. **d** Schematic representation for the role of TRABID targeting in the activation of cGAS/STING pathway to trigger anti-tumor immunity and ICB sensitization. Source data are provided as a Source Data file.

and followed procedures as previously described[72]. Briefly, peptide mixtures were loaded onto a 75 µm ID, 25 cm length PepMap C18 column (Thermo Fisher Scientific) packed with 2 µm particles with a pore width of 100 Å and were separated using a segmented gradient in 120 min from 5% to 35% solvent B (0.1% formic acid in acetonitrile) at a flow rate of 300 nl/min. Solvent A was 0.1% formic acid in water. The mass spectrometer was operated in the data-dependent mode. Survey scans of peptide precursors from 350 to 1600 $m/z$ were performed at 240 K resolution with a $2 \times 10^5$ ion count target. Tandem MS was performed by isolation window at 1.6 Da with the quadrupole, HCD fragmentation with normalized collision energy of 30, and rapid scan MS analysis in the ion trap. The MS$^2$ ion count target was set to $1 \times 10^4$ and the max injection time was 50 ms. Only those precursors with charge state 2–6 were sampled for MS$^2$. The instrument was run in top speed mode with 3 s cycles and the dynamic exclusion duration was set to 15 s with a 10 ppm tolerance around the selected precursor and its isotopes. Monoisotopic precursor selection was turned on.

The raw data were processed for protein identification and SILAC quantification using the MaxQuant software (Version 1.6.15.0, Max Planck Institute of Biochemistry). Protein identification was performed using the Andromeda search engine against the Swiss-prot *Homo sapiens* database (20,376 entries total). Search criteria used were trypsin digestion, allowing up to 2 missed cleavages. The search tolerance parameters were 20 ppm for the first search mass accuracy tolerance, 4.5 ppm for main search mass accuracy, and 0.5 Da for ITMS MS/MS tolerance search. Fixed modifications were set as carbamidomethyl (cysteine) and variable modifications were set as oxidation (methionine) and GlyGly (lysine). Heavy arginine (Arg-10) and lysine (Lys-6) were selected for SILAC quantification. The false discovery rate (FDR) was calculated to 0.01 by the decoy (reverse) database approach.

### ELISA assays
For measuring 2'3'-cGAMP, cells were lysed with NP40 lysis buffer containing 150 mM NaCl, 50 mM Tris-HCl (pH 7.5), 1% NP40, 1% sodium deoxycholate, 1 µg/ml aprotinin, 1 µg/ml leupeptin, and 1 mM PMSF. cGAMP levels in cell lysate were measured by the cGAMP ELISA Kit (Cayman) according to the manufacturer's instructions. For determining IFNβ concentrations, culture supernatants were collected and measured using human IFNβ ELISA Kit (R&D System) according to the manufacturer's instructions. Absorbance was measured at 450 nm (for cGAMP) or 540 nm (for IFNβ) using a microplate reader.

### Luciferase reporter assay
Cells were transfected with pLuc-IFNβ Firefly reporter construct, together with the pTK-Renilla luciferase plasmid for 24 h. Luciferase reporter assay was conducted by the Dual-Luciferase Reporter Assay System (Promega) according to the manufacturer's instructions. The relative promoter activity was expressed as the ratio of Firefly luciferase activity to the Renilla luciferase activity.

### Cell proliferation assay
Cells were seeded on 96-well plates at a density of 1500 cells/well and labeled with 10 µM BrdU for 2 h. BrdU incorporation was determined by BrdU Proliferation Assay Kit (Merck Millipore) according to the manufacturer's instructions.

### Mouse husbandry
Mice were housed in a specific pathogen-free animal facility under temperatures 22 ± 2 °C and humidity-controlled (55 ± 5%) conditions with a 12 h light/12 h dark circadian cycle and access to food and water.

### Animal studies
All mouse experiments were conducted according to the guidelines of animal ethical regulations and approved by the Institutional Animal Care and Use Committee, Academia Sinica, Taiwan. If not, sex was not considered in the study design and the maximum tumor size allowed is 2 cm in diameter. Approximately $5 \times 10^5$ B16F10 cells or their derivatives were injected subcutaneously into the dorsal area of 8 to 10-week-old male C57BL/6 J (C57BL/6NCrlBltw) mice or Nude (Bltw:NU-Foxn1nu) mice purchased from BioLASCo Taiwan Co., Ltd. The size of tumors was determined by a digital caliper and calculated using the modified ellipsoid formula volume = (length × width × height)/2. Tumors were surgically excised on day 15. In the experiments involving antibody and/or inhibitor treatment, B16F10 tumor-bearing mice were randomly divided into four groups. For antibody treatment, mice were given anti-PD-1 antibody (clone 29 F.1A12, Bio X Cell) or an isotype control (clone 2A3, Bio X Cell) at 5 mg/kg body weight via intraperitoneal injection on days 6, 9, and 12 after tumor implantation. For inhibitor treatment, mice were intraperitoneally injected with TRABID inhibitor at a dose of 5 mg/kg body weight on days 6, 9, and 12. For administration of STING inhibitor, mice were intraperitoneally injected with C176 at a dose of 375 mmole in 100 µl corn oil (Sigma) every 3 days.

### Flow cytometry
For analyzing tumor-infiltrating immune cells, tumors were weighed, minced, and incubated with digestion buffer containing HBSS (Sigma-Aldrich), 5 mM CaCl$_2$, 40 U/ml DNase I (BIOTOOLS), 0.5 mg/ml type I collagenase (Thermo Fisher Scientific), 0.5 mg/ml type IV collagenase (Thermo Fisher Scientific), and 0.2 mg/ml hyaluronidase (MedChem-Express) for 20 min at 37 °C. The dissociated tumor cells were passed through a 100 µm cell strainer (BD) and incubated with ammonium chloride potassium (ACK) lysis buffer (MyBioSource). Dead cells were excluded using Fixable Viability Stain 780 (BD). After washing three times with PBS containing 2% inactivated FBS, cell suspensions were blocked with an anti-CD16/32 antibody (BioLegend) and stained with various antibodies to immune cell markers for 30 min on ice, followed by cell fixation and permeabilization using the Transcription Factor Buffer Set (BD) for 1 h on ice and then intracellular staining with the desired antibody for 30 min on ice. Flow cytometry analysis was performed using a Thermo Fisher Scientific Attune NxT Flow Cytometers and analyzed with FlowJo v10.8.1 software. For analyzing cell cycle profiles, DAPI-stained cells were subjected to flow cytometry with a Thermo Fisher Scientific Attune NxT Flow Cytometers together with FlowJo v8 software. For both experiments, Attune NxT software (v 4.2.0) was used for data collection.

### IHC
Tumor sections derived from syngeneic mouse models were prepared by the Pathology Core Facility, Institute of Biomedical Sciences (Academia Sinica, Taiwan). IHC staining was performed with a Novolink Max Polymer kit (Leica Biosystems, Wetzlar, Germany) according to the manufacturer's instructions. Briefly, all slides were dewaxed with xylene/ethanol, and antigen retrieval was performed by boiling in TRS buffer for 10 min. Protein Block was used to cover the slides for 10 min followed by incubation with primary antibodies overnight. The polymer was used for polymerization. The peroxidase activity was visualized with diaminobenzidine tetrahydroxychloride (DAB) solution. The sections were counterstained with hematoxylin (Sigma-Aldrich).

### Bioinformatics
ISG scores were calculated as described[57]. The scores were defined as the mean absolute deviation modified Z-score-normalized mRNA expression data derived from TCGA. Web platform TIMER2.0 (http://timer.comp-genomics.org/) was used for the estimation of the levels of diverse infiltrating immune cells in various cancer types from TCGA. TRABID gene expression was analyzed for the correlation with the abundance of various tumor-infiltrating immune cells. The correlations were visualized as the heatmap with numbers showing the

purity-adjusted Spearman's Rho, together with the P value of statistical significance across various cancer types. For measuring the correlation of the expression of two genes in tumors derived from cancer patients, the expression levels were retrieved from TCGA for Pearson's correlation analysis using the Phyton Statistical package Scipy stats v1.9.0.

## Statistics analysis and reproducibility

Two-tailed, unpaired Student's t-test was used for the comparisons between two groups, and ANOVA was used for multigroup comparisons. Kaplan-Meier estimation and log-rank test were used to compare the mouse survival curves. P value less than 0.05 was considered as statistically significant. All statistical results were derived from three or more biological repeats. For Western blots or confocal images, data are representatives of three (for Fig. 1i, j; Fig. 2d, e, g, h; Fig. 3a–k; Fig. 5a–c, e–g; Supplementary Fig. 1a, b, h–j; Supplementary Fig. 2c, d; Supplementary Fig. 3a–f; Supplementary Fig. 4b; Supplementary Fig. 5f; Supplementary Fig. 7a) or two (for Fig. 2f, i; Fig. 4d, e; Fig. 5d, m, n; Supplementary Fig. 3g; Supplementary Fig. 4a; Supplementary Fig. 5b, e, j; Supplementary Fig. 6b, d; Supplementary Fig. 7d, e) independent experiments.

## Reporting summary

Further information on research design is available in the Nature Portfolio Reporting Summary linked to this article.

## Data availability

The original MS data for comparing proteome changes by TRABID knockdown are deposited to the ProteomeXchange Consortium via PRIDE partner repository with the project accession number PXD035002. The analyzes in Fig. 8a and Supplementary Fig. 8a–q were acquired using TCGA database from online web server UCSC Xena Functional Genomics Explorer (https://xenabrowser.net/), whereas data in Fig. 8b, c were retrieved from TIMER2.0 web platform (http://timer.comp-genomics.org/). The remaining data are available within the article, supplementary information, and source data file. Source data are provided with this paper.

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

## Acknowledgements

We thank Tang K Tang, Li-Chung Hsu, Guang-Chao Chen, Ping-Hung Chen, Che-Ming Jack Hu, Minglei Zhao, and Anthony Kossiakoff for reagents, the Academia Sinica Common Mass Spectrometry Facilities located at the Institute of Biological Chemistry for mass spectrometry analysis, RNA Technology Platforms and Gene Manipulation Core Facility for shRNA constructs, Pathology Core Facility of Institute of Biomedical Sciences, Academia Sinica for tissue section preparation, Flow Cytometry Core Facility of Institute of Biomedical Sciences, Academia Sinica for flow cytometry analysis, and Li-Chun Hsu for discussion and critically reading the manuscript. This work was supported by the Ministry of Science and Technology Academic Summit Grant (MOST 108-2639-B-001-ASP to R.-H.C.), an intramural fund from the Institute of Biological Chemistry, Academia Sinica (to R.-H.C.), and the Ministry of Science and Technology Grants (MOST 110-2628-B-A49A-508 and MOST 110-2326-B-A49A-503-MY3 to W.-J.W.). Y.-H.C. is supported by Postdoctoral Research Fellowship, Academia Sinica.

## Author contributions

Y.-H.C. and H.-H.C. conceived the study, designed and performed most experiments, and analyzed the data. W.-J.W., W.-S.H., and C.-H.K. designed and performed the mitotic cell division analyzes. H.-Y.C. performed bioinformatics and FISH analyzes. S.-R.L. assisted in some

deubiquitination analyzes. N.Y.Y. performed histopathological analysis. R.-L.Y. performed PLA analysis. S.-J.C. helped with some animal studies. K.-P.W. helped with protein purification. R.-H.C. directed and coordinated the study, designed the research, and oversaw the project. R.-H.C. and Y.-H.C. wrote the manuscript.

## Competing interests

The authors declare no competing interests.
