## [Peer Review File · Nature Communications]

TRABID inhibition activates cGAS/STING-mediated anti-tumor immunity through mitosis and autophagy dysregulationREVIEWER COMMENTS

Reviewer #1 (Remarks to the Author): with expertise in TRABID, ubiquitination

The context for this research is particularly important given recent evidence indicating that activation of innate immunity pathways like cGAS/STING can improve the efficacy of immune checkpoint blockade therapies. This manuscript by Chen YH and colleagues aims to implicate the deubiquitinase TRABID in cGAS/STING signalling for the first time and proposes TRABID inhibition as a way to sensitize cells to improve the efficacy of ICB. This is an exciting proposition which would establish cross talks between ubiquitin signalling and innate immune pathway and could lead to improved therapies.

A number of studies have proposed TRABID exhibits pro-tumor effects, and previous data from mice studies have shown that TRABID deubiquitylase activity epigenetically regulates interleukin 12 and 23. Although a link between TRABID and innate immune response has been proposed previously in *Drosophila*, this is the further report to directly implicate TRABID in cGAS/STING signalling and this is very exciting indeed.

Having clearly stated their hypothesis at the onset, it is not clear to me why the authors decided to explore the role of TRABID on mitosis and perhaps this could be better explained as part of the narrative. Further, the authors previously reported a novel function for TRABID as a positive regulator of autophagy while the E3L UBE3C was shown to antagonise autophagy, through addition of branched K29/K48-linked chains on the class III PI3-kinase VPS34. It is also not entirely clear how the authors went from these series of observations to start focusing on the cGAS/STING pathway and mitosis and this could perhaps be better conveyed.

Some recent studies on TRABID and K29 ubiquitin biology would benefit from being included as references to further back up the claims made in this manuscript. Specifically, Yu Y and colleagues *Nat Chem Biol* 17, pages 896–905 (2021) showed overexpression of wild-type and active TRABID led to a G1 block. This study went on to use the first Ubiquitin K29 affimer which revealed that the K29 ubiquitin linkages are preferentially recognised and processed by TRABID were abundant during mitosis in particular around the midbody during telophase. Harris et al *JBC* 2020 established HECTD1 as a one possible partner E3 of TRABID and proposed that this DUB/E3 pair is specific for ubiquitin chains assembled via K29. A subsequent study on HECTD1 revealed this E3 contributes to cell proliferation through its ubiquitin ligase activity, specifically during mitosis. Together these studies add weight and complement the observations made by Chen YH and colleagues that TRABID is important in mitosis.

The authors indeed provide compelling evidence that TRABID depletion (KO or shRNA) triggers mitotic defect including chromosomal abnormalities and micronuclei (Fig 1). To add strength to these observations and the direct nature of TRABID effect on mitosis, the authors should try and rescue some of the phenotypes observed using either wild-type or mutant TRABID. Given the point mutant/catalytic-dead TRABID C443S traps ubiquitin and can exert dominant negative effect, I would suggest using the 3xNZF/C443S mutant since this mutant cannot bind nor cleave ubiquitin chains. Furthermore I would suggest the authors also test whether the loss of function data on TRABID can be recapitulated in a transient siRNA system as this would also provide more assurance of the direct nature of the effect.

Interestingly Vaughan et al *Sci Rep* 2022 recently established that HECTD1-depleted cells take longer to go from Nuclear Envelope Breakdown to anaphase onset, and although this is not observed in the TRABID KO MEFs, TRABID-depleted cells still exhibit increased metaphase to mitotic exit. Could the authors explain the difference in these two mitotic readout and provide some explanation as to why no change was observed for NEBD-anaphase onset vs. metaphase to mitotic exit?

Given Harris et al *JBC* 2021 established TRABID/HECTD1 as a DUB/E3 pair for K29 linkages, this new data by Chen YH et al on TRABID further strengthen the idea that K29 chains are functional

signals during mitosis. In fact, Yu Y et al Nat Chem manuscript (Figure 6) revealed that K29 linkages are indeed detected throughout mitosis which again adds weight to role of K29 ubiquitin and the enzymes which regulate this modification during mitotic progression.

The authors went on to explore TRABID function during mitosis and identify candidate substrates using SILAC proteomics. Protein levels of components of the chromosomal passenger complex (CPC) including Aurora B and survivin were reduced in TRABID-depleted HeLa cells and the levels were stabilised by MG132 treatment. To add support to this, I would suggest the authors try and rescue Aurora B and Survivin levels using wild vs. mutant TRABID as mentioned previously. Could the authors also explain why they decided to look for interactors in HeLa cells while the bulk of the phenotypic data provided in figure 1 was done using MEF WT vs KO? The finding that TRABID protein levels increases during mitosis (nocodazole treated cells) is particularly interesting and so is the PLA which shows interaction between Aurora B and TRABID in mitotic cells at anaphase. Have the authors tried to recapitulate these PLA experiments by staining for Aurora B and K29 Vs K63 ubiquitin using the published affimer, perhaps even using the Catalytic trapping C443S mutant? This could help establish the type of ubiquitin chains TRABID processes during mitosis.

In Fig 2G the authors show interesting data that HECTD1 is a novel CPC component although it is not clear to me why it was necessary to inhibit the proteasome for those interaction studies? Please comment. In addition and to add further weight to this, it would be important to determine how TRABID is recruited to this complex, via its NZFs or OTU domain? Although TRABID deubiquitinase activity has been linked directly to the cell cycle, through K29 ubiquitin chains, this exciting report by Chen YH et al is the first to directly implicate TRABID in mitosis.

The next obvious line of investigation the authors followed was to establish that TRABID regulates the ubiquitination status of interacting partners Aurora B and surviving. Unfortunately this figure is a major weakness of the paper as the strategy used induces background which can be misinterpreted for protein ubiquitination. For instance, the GFP and FLAG WB following His-Ubiquitin pulldown by Ni-NTA yield bands corresponding to the unmodified version of Aurora B and survivin. Therefore some of the bands in these smear likely represent background rather than protein polyubiquitination, and the similarly in the pattern of the smear for the different protein tested is further indication of this. Another concerning results is the fact that the TRABID C/S does not really traps more ubiquitin and this is counter intuitive to what many studies have shown. As an alternative approach to try and establish the fact that Aurora B and surviving are substrate of TRABID DUB activity, I would suggest the authors use TRABID NZFs or the recently reported K29 affimer in pulldown or IP experiments.

Data presented in Fig 3F and G which aim to show that addition of recombinant wild-type TRABID reduces ubiquitylation of GFP-Aurora B and Flag-Survivin, is also not convincing. For instance the 100Kda band on Fig 3F would correspond to GFP-Aurora B with at least 5-6 ubiquitin moiety and this does not change. Using ubiquitin binding domain of TRABID (and potentially other Tandem Ubiquitin Binding Entities specific for different linkages, TUBES) or the K29 affimer in pulldown assay would overcome this since and would also help to directly implicate the ubiquitin chain type involved.

It also seems that much of the signal detected on Fig 3G does not represent ubiquitination event since the lowest band corresponds to unmodified Flag-Survivin can be detected following the Ni-NTA IP, therefore casting doubt as to which signal/band correspond to protein ubiquitination. The band migrating at 25kDa could indicate mono ubiquitination of Survivin but based on the data provided but I am not convinced there is evidence for polyubiquitination of Survivin. Same comment for GFP-Aurora B in panel 3F and other panel too, where the non-modified version for the protein seems to be detected following Ni-NTA IP which also indicates background. Some of these experiments would benefit from being carried out with a K0 ubiquitin mutant to firmly establish which signal represent polyubiquitination VS. multi monoubiquitination.

Figure 3i: I am puzzled by the fact that MG132 rescue levels of the unmodified GFP-Aurora B and FLAG-Survivin. Could the author comment on why the unmodified and not the ubiquitinated pool of these proteins is detected in MG132 treated cells. Presumably MG132 blocks polyubiquitinated

proteins from being degraded, and non the non-polyubiquitinated ones?

In figure 4, the authors provide evidence that TRABID deletion increases the proportion of micronuclei. Although this looks compelling, I am sceptical about the use of the small molecule TRABID inhibitor used in these studies since this compound has not been fully characterised including towards full-length TRABID DUB activity or other OTU DUBs, or its DUB activity towards K29 linkages specifically. There is also no proteomics or RNA-Seq data to show the specificity of this compound, what else does it affect? Please comment. Nevertheless the data showing that VPS34, Aurora B and Survivin can rescue the micronuclei phenotype produced by TRABID depletion support the hypothesis, Fig 4c.

The recent evidence that cGAS acts as a receptor to sense micronuclei and trigger micronucleophagy as well as cGAS autophagic degradation is a very interesting narrative to put the data obtained by the authors in context (Fig 4 and 5). The author rightfully explored the possibility that the reduced micronucleophagy observed upon TRABID depletion might stabilise cGAS. However, I find it difficult to reach the same conclusion given the overexposure of the GAPDH and the weak reduction in cGAS observed in Fig 5A. This data is not convincing enough to make the claim that TRABID also impacts on cGAS protein levels, in the absence of a rescue experiment. I also do not see an increase in p-STING in Fig 5E, given that the change observed for p-STING is matched in the FLAG WB (which seems overexposed). However, TRABID depletion seems to clearly have an effect on type 1 interferon genes and ISGs mRNA expression, although the authors should provide a clearer and more details explanation to try and reconcile this with the data from Jin J et al Nat Immunology who showed where IL12 and 23 levels were reduced upon TRABID deletion in mice. Please explain.

Although the transcriptomics data looks exciting, the literature has established a low correlation between mRNA expression and protein translation. Have the authors measured proteins levels of some of the IFNs, in particular PD-L1, for example by ELISA. This seems essential to make a more direct link here between the increase in PD-L1 upon TRABID depletion and the increase response to ICB.

The effect of TRABID depletion on reducing tumour growth and increasing a anti-tumor microenvironment is also convincing, but again a rescue experiment should be included to establish direct implication of TRABID DUB activity (Fig 6).

Figure 7 aims to establish that loss of TRABID function could improve efficacy of ICB through upregulation of PD-L1. This experiment set up by the authors is the correct one to test their hypothesis but I must reiterate my concerns with regards to the poorly characterised TRABID inhibitor used. This is in my mind is a major concern as there is not enough validation of this inhibitor, even in the original paper referenced, to bring confidence that this compound specifically target TRABID and not any other OTU DUB or the possibility that it could induce pleiotropic effect. The data look promising and unfortunately at this stage I am not aware of another publicly available TRABID inhibitor.

It is not clear to me whether the authors are proposing that TRABID depletion reduces cGAS during mitosis or another cell cycle stage. Please comment.

There is a lot of effects suggested to be driven by TRABID depletion and the narrative of the authors aims to place these different functions into a logical, linear pathway. Although I agree with the effect of TRABID depletion and the interaction with CPC, I am less convinced by the function of TRABID DUB activity in that process given the ubiquitination assay which I see as inconclusive and also the use of the TRABID inhibitor which has not been fully evaluated and validated. The effect of TRABID depletion on micronuclei and the induction of IFNs and ISGC mRNA transcription is also convincing but it is rather difficult to confidently put all these cellular processes into a linear pathway given the data provided.

Reviewer #2 (Remarks to the Author): with expertise in chromosomal instability, mitosis, cancer, STING

In their manuscript 'TRABID inhibition activates cGAS/STING-mediated anti-tumor immunity through mitosis and autophagy dysregulation', Chen and co-workers have studied the effects of depletion of inhibition of TRABID, more commonly referred to as ZRANB1.

The authors show a number of interesting phenotypes, and show that TRABID can act as a DUB for aurora B and survivin. Also, they show that inactivation of TRABID leads to inflammatory signaling, and sensitivity of traid inhibited cells to ICI agents.

Overall, some of the phenotypes are impressive, but I feel that the observations remain poorly understood at the mechanistic level.

Specific comments:

- Line 6: 'Since the DNA-sensing ability of cGAS is independent of DNA source and sequence, self DNAs accumulated in the cytoplasm enable the activation of cGAS/STING pathway'. Increasingly, the structure of extranuclear DNA is recognized to determine its ability to trigger cGAS. Actually, the chromatin context determines cgas activation, for instance PMID: 32913000

-mitotic exit is poorly defined. In the upper example of the KO cells, there is no anaphase, so anaphase onset cannot be used.

-Is it surprising that 1 in 5 control cells does not have a bipolar spindle. I realize that cells in culture have a background level of chromosome missegrations, but this is unexpectedly high.

-to link the phenotype of increased micronuclei to an underlying mechanism, it is very important to test whether they are centromere positive

-Figure 2B: the color coding is difficult to interpret. Please use bargraphs or another way of displaying that is more quantitatively interpretable.

-Figure 2A/B: traid does many things (as the authors also discuss in their introduction), including regulating transcription factors that are involved in proliferation. All the highlighted proteins are cell cycle regulated. Can a GSEA-type approach be used on the MS dataset to see if these proteins are part of a proliferation signature?

Figure 2C/D: many of the observed effects can be explained by traid-depleted cells to proliferate slower. This should be checked. Also, the treatments in Figure 2C/D should be accompanied by flow cytometry analysis (DNA, plus mitotic marker).

Figure 3: These data show that overexpressed traid can deubiquitlate overexpressed survivin and aurora B in vitro, and that depletion of traid leads to slightly elevated levels of ubiquitylated versions of overexpressed survivin and aurora B. the authors should also show that endogenous survivin or aurora B are more ubiquitylated in the absence of traid.

Also, it appears that traid depletion can lead to survivin or aurora B ubiquitylation in nocodazole, a situation when APC/C-Cdh1 is not active yet due to high CDK activity. This is highly unexpected. How do the authors explain these data?

Figure 4: This figure needs western blots and IF to confirm that the levels of aurora B, survivin and VPS34 are expressed to control levels. Also, transient transfection usually does not give a 100% transfection efficiency, although microscopy is used to analyse all cells? How do the authors know if all cells that are analyzed contain all the transiently expressed constructs? Same applies to figure 5i.

Figure 5a/b: TRAbid aurora B and blots should be included to control for knockdown.

Figure 5: ‘..while Aurora B depletion reduced cGAS abundance through a lysosome-dependent mechanism’. This could indeed be, but there are so many controls missing here, that is a wild overinterpretation of the results.

Figure 5: the authors should show that the effects are dependent on cGAS/STING, through siRNA or knockouts of these genes.

Figure 6: the TRABID-depleted tumors are smaller and grow less fast. This could indicate strong defects in proliferation, that also affect all other assays in the manuscript. The authors should check for proliferation rates (ki67 positivity in tumors appears very different!).

The manuscript needs extensive grammar checks. Few examples below:

Line 29 prevents -> protects

Line 42: show -> shows

Line 44: of -> to

Line 48: ‘for’ should be left out

Line 60: ‘are testing’ -> have been tested?

Line 132: ‘no’ -> not

Reviewer #3 (Remarks to the Author): with expertise in autophagy, cGAS/STING

In this study, Chen YH et al examined a novel function of TRABID, a DUB in regulating cGAS/STING-mediated anti-tumor immunity via altered mitosis and autophagy. First, they established the role of TRABID in mitosis by using TRABID-KO MEFs and other cells. Second, they found that TRABID is upregulated in mitosis and that TRABID regulates mitosis through deubiquitinating and stabilizing CPC such as Aurora B and Survivin. Third, TRABID deficiency causes micronuclei through a combinatory defect in mitosis and autophagy. Fourth, with the increase level of cytosolic micronuclei caused by TRABID deficiency, the cGAS/STING pathway is activated that contributes to the tumor-suppressive effect of Trabid deficiency. Fifth, using animal models, the authors provided evidence that inhibition of TRABID sensitizes tumor response to anti-PD-1 therapy. Finally, TRABID expression in most solid cancer types correlates inversely with an interferon signature and infiltration of anti-tumor immune cells. Overall, this is a very comprehensive study with convincing data, covering cell, animal and clinical aspects with deep mechanistical understanding and potential application in cancer immunotherapy.

There are several points to be addressed to improve the MS.

1) In the mechanistic study, the authors presented two parallel pathways: mitosis defects and impaired autophagy, without establishing the possible link btw these two processes. Based on the knowledge that autophagy is also implicated in regulation of cell cycle esp the M phase (PMID 31733992, 30898011), it would be important for the authors to address this possible link, or at least discuss this point.

2) In the in vitro experiments using cell lines, the authors only used siRNA knockdown of TRABID in HeLa cells. The study will be significantly enhanced if they can establish the TRABID KO stable cells using CRISPR (in addition to the MEFs), plus the reconstitution of both the WT and enzyme-activity dead mutant, to replicate some of the key observations such as changes of Aurora B and Survivin protein level, formation of cytosolic micronuclei, etc.

3) One key point of this study is that TRABID deubiquitinates and stabilizes Aurora B and Survivin. However, as stated by the authors, TRABID is a DUB that specifically disassembles K29- and K33-linked ubiquitin chains. Then, the authors need more evidence to highlight that how K29- and K33 Ub are implicated in regulating the protein stability of Aurora B and Survivin. Will MG132 prevent the reduction of protein levels of CPC components in shTRABD cells treated with nocodazole (Figure

2e)?

Other points:

- 1) How does nocodazole treatment (cells in mitosis) enhance the expression of TRABID (both mRNA and protein)?
- 2) Figure 8d: In this illustration, the link between cGAS and micronuclei is not clearly illustrated.

Reviewer #4 (Remarks to the Author): with expertise in cGAS/STING

This manuscript reports a series of previously unrecognized consequence of lacking the ubiquitin thioesterase, TRABID, aka ZRANB1, in cells. First, the authors show that TRABID deficiency caused chromatin mis-segregation and cytokinesis failure. Such mitotic errors lead to more micronuclei, thus activate the cGAS-STING pathway. Then they applied quantitative proteomics to show that a few proteins are downregulated upon TRABID knockdown, including Aurora Kinase B and Survivin, which play critical roles in mitotic chromatin segregation. Plus, they demonstrate that TRABID can de-ubiquitinate Aurora Kinase B and Survivin. Because TRABID deficiency engages the innate immune response, the authors went further to show that removing TRABID in tumor cells can elicit anti-tumor response and TRABID inhibitor can improve the anti-tumor effect of immune checkpoint blockade by the anti-PD-L1 antibody. Together, this is a quite comprehensive study that report some interesting findings on TRABID. However, the authors must address the following major and minor concerns.

1. TRABID specifically bind and remove K29-linked and K33-linked ubiquitin. Are Aurora B and Survivin modified by these special forms of ubiquitin chains?
2. TRABID regulates many different proteins via de-ubiquitination, as the SILAC experiments show. How confident are you that the effect on cGAS dysregulation is mainly via Aurora Kinase B and VPS34? In those rescue experiments with Aurora B and VPS34 overexpression, it is necessary to show by immunoblotting that levels of these proteins are restored to normal levels.
3. Page 13 Fig Sfig 5. cGAS inhibitor G140 can reduce interferon in TRABID knockdown cells. It would be more convincing to show similar experiments with STING inhibitor H151/C176, and with specific knockout/knockdown of cGAS or STING.
4. Fig 3a, second lane: as there was no his-Ub, how come there was GFP-Aurora B in the IP? Fig 3b, what is the 70kDa band in Lane 2 of the anti-his immunoblot?
5. Fig 3f-g showed that purified his-TRABID can de-ubiquitinate Aurora Kinase B and Survivin in vitro. The concern is whether the activity comes from his-TRABID or co-purified contaminant(s). Please show Coomassie gel of his-TRABID to show protein purity. Also consider use catalytic dead mutant of TRABID as a negative control.
6. Page 3 Line 53 "cGAS binds and activates STING..." cGAS does not bind STING. It activates STING via its product, second messenger, cGAMP.
7. Fig 5a, why would Aurora Kinase B knockdown cause reduction in cGAS level via lysosomal-dependent mechanism? Aurora Kinase B knockdown will cause mitotic errors, like TRABID knockdown does, and it will likely induce cGAS activation via micronuclei. This may lead to increased lysosomal degradation of micronuclei. However, not all cGAS is in micronuclei, and elevated interferon may also induce more cGAS (as an ISG). Thus, it is surprising that AurkB knockdown cause substantial cGAS reduction.
8. Fig 6a. TRABID knockdown retarded tumor growth of B16F10 cells. One possible explanation is that TRABID knockdown may reduce the fitness of B16F10 through compromising cell cycle progression, which by itself would be sufficient to reduce tumorigenesis. How much does the anti-tumor immunity contribute to this reduced tumorigenesis is unknown.
9. Aurora Kinase B was recently reported to phosphorylate cGAS during cell cycle, and hyperphosphorylation inhibits cGAS activity. Conceptually, this connection might play a role in the immune outcome of TRABID deficiency. Should include this in the discussion.

Point-by-point responses to reviewers' comments

Reviewer #1	pp. 1-13
Reviewer #2	pp. 14-19
Reviewer #3	pp. 20-22
Reviewer #4	pp. 23-26

Reviewer #1 (Remarks to the Author): with expertise in TRABID, ubiquitination

The context for this research is particularly important given recent evidence indicating that activation of innate immunity pathways like cGAS/STING can improve the efficacy of immune checkpoint blockade therapies. This manuscript by Chen YH and colleagues aims to implicate the deubiquitinase TRABID in cGAS/STING signalling for the first time and proposes TRABID inhibition as a way to sensitize cells to improve the efficacy of ICB. This is an exciting proposition which would establish cross talks between ubiquitin signalling and innate immune pathway and could lead to improved therapies.

A number of studies have proposed TRABID exhibits pro-tumor effects, and previous data from mice studies have shown that TRABID deubiquitylase activity epigenetically regulates interleukin 12 and 23. Although a link between TRABID and innate immune response has been proposed previously in Drosophila, this is the further report to directly implicate TRABID in cGAS/STING signalling and this is very exciting indeed.

Ans: We appreciate this reviewer for supporting the importance of our studies.

Having clearly stated their hypothesis at the onset, it is not clear to me why the authors decided to explore the role of TRABID on mitosis and perhaps this could be better explained as part of the narrative. Further, the authors previously reported a novel function for TRABID as a positive regulator of autophagy while the E3L UBE3C was shown to antagonise autophagy, through addition of branched K29/K48-linked chains on the class III PI3-kinase VPS34. it is also not entirely clear how the authors went from these series of observations to start focusing on the cGAS/STING pathway and mitosis and this could perhaps be better conveyed.

Ans: In fact, this study was initiated entirely from the observation of morphologic features of Trabid KO MEFs. We established such MEFs for other purpose but accidentally found morphological abnormalities such as enlarged nuclear areas and micronuclei. Inspection of cell morphology is a routine for cell culture and does not

require a particular rationale. We did mention in the manuscript (Lines 129-131) that such morphological abnormalities suggest chromosome instability and chromosome number alterations, both of which are linked to mitosis defects. In the revised manuscript, we provided new data showing that MEFs deficient in a key autophagy gene Atg5 do not display such abnormalities (Supplementary Fig. 1e), thus suggesting the existence of an autophagy-independent function of Trabid. As to the rationale for studying cGAS/STING pathway, we mentioned in Lines 260-270 that “the cytosolic micronuclei are sensed by cGAS, leading to the activation of cGAS/STING pathway”. Since this is a well-known process, a number of references are cited.

Some recent studies on TRABID and K29 ubiquitin biology would benefit from being included as references to further back up the claims made in this manuscript. Specifically, Yu Y and colleagues Nat Chem Biol 17, pages 896–905 (2021) showed overexpression of wild-type and active TRABID led to a G1 block. This study went on to use the first Ubiquitin K29 affimer which revealed that the K29 ubiquitin linkages are preferentially recognised and processed by TRABID were abundant during mitosis in particular around the midbody during telophase. Harris et al JBC 2020 established HECTD1 as a one possible partner E3 of TRABID and proposed that this DUB/E3 pair is specific for ubiquitin chains assembled via K29. A subsequent study on HECTD1 revealed this E3 contributes to cell proliferation through its ubiquitin ligase activity, specifically during mitosis. Together these studies add weight and complement the observations made by Chen YH and colleagues that TRABID is important in mitosis.

Ans: We appreciate the reviewer for these valuable comments. We added one paragraph in the Discussion section (Lines 540-550) for comparing our study with these previous studies for the cell cycle functions of K29-linked polyubiquitination.

The authors indeed provide compelling evidence that TRABID depletion (KO or shRNA) triggers mitotic defect including chromosomal abnormalities and micronuclei (Fig 1). To add strength to these observations and the direct nature of TRABID effect on mitosis, the authors should try and rescue some of the phenotypes observed using either wild-type or mutant TRABID. Given the point mutant/catalytic-dead TRABID C443S traps ubiquitin and can exert dominant negative effect, I would suggest using the 3xNZF/C443S mutant since this mutant cannot bind nor cleave ubiquitin chains. Furthermore I would suggest the authors also test whether the loss of function data on TRABID can be recapitulated in a transient siRNA system as this would also provide more assurance of the direct nature of the effect.

Ans: In revised manuscript, we added new data showing that transient knockdown of

TRABID by siRNA increases mitotic errors (Fig. 1i). In addition, the mitotic errors caused by TRABID siRNA are rescued by re-expression of TRABID wild type but not TRABID Δ NZF1/CS mutant (called Δ N/CS throughout the manuscript). (Fig. 1j). Notably, we chose the deletion of NZF1 in combined with mutation of the catalytic residue is due to the specificity of NZF1, but not NZF2 or NZF3, in binding to K29- and K33-linked Ub chains. Accordingly, mutation of NZF1, but not NZF2 or NZF3, disrupts the distribution of TRABID to Ub-enriched puncta (*Mol Cell* 58: 95-109. 2015). With the inclusion of transient knockdown data and rescue data, we believe that our study has provided compelling evidence for a key role of TRABID in mitotic progression through its catalytic and Ub chain-binding activities.

Interestingly Vaughan et al Sci Rep 2022 recently established that HECTD1-depleted cells take longer to go from Nuclear Envelope Breakdown to anaphase onset, and although this is not observed in the TRABID KO MEFs, TRABID-depleted cells still exhibit increased metaphase to mitotic exit. Could the authors explain the difference in these two mitotic readout and provide some explanation as to why no change was observed for NEBD-anaphase onset vs. metaphase to mitotic exit?

Ans: We do not see any conflict between our study and the study by Vaughan et al. Since HECTD1 and TRABID are E3 ligase-DUB pair, they should not elicit the same functions, even if they act on the same substrate. Furthermore, while our study indicates that **K29-deubiquitination** of Aurora B and Survivin is needed for cell cycle to progress through late mitosis (from metaphase to mitosis exit), the paper by Vaughan et al. indicates that **K29/K48-ubiquitination** of an unidentified protein is needed for cell cycle to progress through the early mitosis (from NEBD to anaphase). In this previous study, although the candidate protein undergoing K29/K48-ubiquitination has not been rigorously characterized, mitosis checkpoint complex was implicated. It is well conceivable that one protein whose K29/K48-ubiquitination is required for the early phase of mitosis, whereas other proteins (i.e., Aurora B and Survivin) whose K29-deubiquitination is involved in the late phase of mitosis. There is no any conflict. Additionally, we would like to point out that HECTD1 is not the only E3 ligase that can antagonize the function of TRABID. UBE3C is also capable of assembling the K29/K48 branched Ub chain (*JBC* 276: 19871-8, 2001; *Nature Commun* 12: 1322, 2021) and can antagonize the functions of TRABID (*Nature Commun* 12: 1322, 2021; *JBC* 296: 100246, 2021). Thus, we cannot deduce that HECTD1 is the E3 ligase that antagonizes the functions of TRABID on Aurora B/Survivin modification.

Given Harris et al JBC 2021 established TRABID/HECTD1 as a DUB/E3 pair for

K29 linkages, this new data by Chen YH et al on TRABID further strengthen the idea that K29 chains are functional signals during mitosis. In fact, Yu Y et al Nat Chem manuscript (Figure 6) revealed that K29 linkages are indeed detected throughout mitosis which again adds weight to role of K29 ubiquitin and the enzymes which regulate this modification during mitotic progression.

Ans: As mentioned earlier, we added one paragraph (Lines 540-550) to discuss these issues. In addition, although the K29-ubiquitination signal was detected throughout the mitosis period, it is not particularly abundant in mitosis (see Extended Data Fig. 9a of the *Nat Chem Biol* paper).

The authors went on to explore TRABID function during mitosis and identify candidate substrates using SILAC proteomics. Protein levels of components of the chromosomal passenger complex (CPC) including Aurora B and survivin were reduced in TRABID-depleted HeLa cells and the levels were stabilised by MG132 treatment. To add support to this, I would suggest the authors try and rescue Aurora B and Survivin levels using wild vs. mutant TRABID as mentioned previously. Could the authors also explain why they decided to look for interactors in HeLa cells while the bulk of the phenotypic data provided in figure 1 was done using MEF WT vs KO? The finding that TRABID protein levels increases during mitosis (nocodazole treated cells) is particularly interesting and so is the PLA which shows interaction between Aurora B and TRABID in mitotic cells at anaphase. Have the authors tried to recapitulate these PLA experiments by staining for Aurora B and K29 Vs K63 ubiquitin using the published affimer, perhaps even using the Catalytic trapping C443S mutant? This could help establish the type of ubiquitin chains TRABID processes during mitosis.

Ans: (1) We provided new data in Fig. 2f showing that re-expression of TRABID WT, but not TRABID Δ N/CS mutant, in TRABID knockdown cells increases Aurora B and survivin abundance. (2) Regarding to the utilization of HeLa cells for interactome analysis and MEFs for mitosis analysis, we agree with the reviewer that it seems a bit incongruous. However, the interactome analysis was in fact did prior to the observation of morphological abnormalities of Travid KO MEFs for a resource basis. Furthermore, this caveat was eliminated in the revised manuscript by including a large amount of mitosis data in HeLa cells, such as TRABID knockdown by shRNA (Fig. 1h), TRABID knockdown by transient expression of siRNA (Fig. 1i), and rescue experiment by re-introducing TRABID WT and Δ N/CS mutant in TRABID siRNA-expressing cells (Fig. 1j). Thus, even though the effect of TRABID on mitotic cell division was originally observed in MEFs, we did confirm this effect in HeLa cells with rigorous studies. (3) Regarding to the PLA assay using K29 affimer, it is not technically feasible, as we could not obtain a PLA kit with F(ab')₂ fragment as the

Plus or Minus PLA probe. Nevertheless, the colocalization of CPC components Aurora B and INCENP with K29 Ub chain in midbody during telophase was already published in the *Nat Chem Biol* paper by immunofluorescence staining (Extended Data Fig. 9f and 10d). We therefore do not need to repeat such experiment.

In Fig 2G the authors show interesting data that HECTD1 is a novel CPC component although it is not clear to me why it was necessary to inhibit the proteasome for those interaction studies? Please comment. In addition and to add further weight to this, it would be important to determine how TRABID is recruited to this complex, via its NZFs or OTU domain? Although TRABID deubiquitinase activity has been linked directly to the cell cycle, through K29 ubiquitin chains, this exciting report by Chen YH et al is the first to directly implicate TRABID in mitosis.

Ans: (1) We agree with the reviewer that there is no need to include MG132 treatment in the experiment showing that TRABID interacts with CPC in mitosis. We therefore replaced this piece of data with a new version without MG132 treatment, in which the interactions are still detected (Fig. 2h). (2) Regarding to whether NZF or OTU domain of TRABID is capable of interacting with CPC, previous studies reported that NZF domain is responsible for binding substrates (*JBC* 296: 100246, 2021; *Mol Cell* 58: 95-109, 2015). However, since CPC is known to form condensates in vitro and in vivo (*Nature Cell Biol.* 21: 1127-37, 2019), whether a clear-cut result can be obtained for Aurora B and Survivin is uncertain. One possibility is that a fraction of CPC that is pulled down by TRABID antibody is due to their interactions with other CPC molecules and/or other components in the condensates. Based on the confirmation basis of the study and the potential caveat, we decided not to perform such experiment.

The next obvious line of investigation the authors followed was to establish that TRABID regulates the ubiquitination status of interacting partners Aurora B and surviving. Unfortunately this figure is a major weakness of the paper as the strategy used induces background which can be misinterpreted for protein ubiquitination. For instance, the GFP and FLAG WB following His-Ubiquitin pulldown by Ni-NTA yield bands corresponding to the unmodified version of Aurora B and survivin. Therefore some of the bands in these smear likely represent background rather than protein polyubiquitination, and the similarly in the pattern of the smear for the different protein tested is further indication of this. Another concerning results is the fact that the TRABID C/S does not really traps more ubiquitin and this is counter intuitive to what many studies have shown. As an alternative approach to try and establish the fact that Aurora B and surviving are substrate of TRABID DUB activity, I would

suggest the authors use TRABID NZFs or the recently reported K29 affimer in pulldown or IP experiments.

Ans: We appreciate the reviewer for pointing out the weaknesses of our data on deubiquitination analyses. In revised manuscript, we followed reviewer's suggestions to have significantly improved data quality and included many pieces of new data. First, we used TRABID Δ N/CS mutant, instead of CS mutant, for the studies shown in Fig. 3a, b to avoid the complication of a dominant negative effect. Second, we used more stringent wash conditions so that nonspecific and unmodified species can no longer be seen (Fig. 3a-d). Third, we provided an additional set of data using the K29 chain-specific binder (sAB-K29) suggested by the reviewer to monitor the K29-ubiquitination of Aurora B and Survivin. Our findings indicate the decrease of K29-linked ubiquitination of Aurora B and Survivin by TRABID overexpression, whereas TRABID knockdown showed an opposite effect (Supplementary Fig. 3c-f). Importantly, sAB-K29 did not detect unmodified substrates. Finally, as requested by reviewer #2, we provided new data showing the ability of TRABID to decrease the K29-linked ubiquitination on endogenous Aurora B and Survivin (Fig. 3e, f). With this large set of new and improved data, we believe that we have provided substantial and convincing evidence for the role of TRABID in promoting the K29-deubiquitination of Aurora B and Survivin.

Data presented in Fig 3F and G which aim to show that addition of recombinant wild-type TRABID reduces ubiquitylation of GFP-Aurora B and Flag-Survivin, is also not convincing. For instance the 100Kda band on Fig 3F would correspond to GFP-Aurora B with at least 5-6 ubiquitin moiety and this does not change. Using ubiquitin binding domain of TRABID (and potentially other Tandem Ubiquitin Binding Entities specific for different linkages, TUBES) or the K29 affimer in pulldown assay would overcome this since and would also help to directly implicate the ubiquitin chain type involved.

It also seems that much of the signal detected on Fig 3G does not represent ubiquitination event since the lowest band corresponds to unmodified Flag-Survivin can be detected following the NI-NTA IP, therefore casting doubt as to which signal/band correspond to protein ubiquitination. The band migrating at 25kDA could indicate mono ubiquitination of Survivin but based on the data provided but I am not convinced there is evidence for polyubiquitination of Survivin. Same comment for GFP-Aurora B in panel 3F and other panel too, where the non-modified version for the protein seems to be detected following Ni-NTA IP which also indicates background. Some of these experiments would benefit from being carried out with a KO ubiquitin mutant to firmly establish which signal represent polyubiquitination VS.

multi monoubiquitination.

Ans: Again, the reviewer pointed out a number of problems for the data of in vitro deubiquitination assays. We therefore replaced the two figures with new data, in which we followed reviewer's suggestion to use sAB-K29 for eliminating the nonspecific problems and, in the meantime, revealing the chain type information. As shown in Fig. 3h, i, recombinant TRABID was able to reduce sAB-K29 signals throughout the lanes and no unmodified substrates could be seen.

Figure 3i: I am puzzled by the fact that MG132 rescue levels of the unmodified GFP-Aurora B and FLAG-Survivin. Could the author comment on why the unmodified and not the ubiquitinated pool of these proteins is detected in MG132 treated cells. Presumably MG132 blocks polyubiquitinated proteins from being degraded, and not the non-polyubiquitinated ones?

Ans: It is conceivable that blockage of the degradation of polyubiquitinated proteins leads to the increase of their unmodified version in vivo, because DUBs in cells could readily remove the ubiquitin chain to result in the unmodified proteins. This effect of MG132 is very commonly observed for ubiquitinated proteins undergoing a proteasomal degradation fate, including a number of TRABID substrates (*Nature Immunol* 17: 259, 2016, Fig. 6g; *Cell Rep* 23: 823, 2018, Fig. 6H; *Nature Commun* 12:1322, 2021, Fig. 2m; *Am J Cancer Res* 11:4807, 2021, Fig. 7C).

In figure 4, the authors provide evidence that TRABID deletion increases the proportion of micronuclei. Although this looks compelling, I am sceptical about the use of the small molecule TRABID inhibitor used in these studies since this compound has not been fully characterised including towards full-length TRABID DUB activity or other OTU DUBs, or its DUB activity towards K29 linkages specifically. There is also no proteomics or RNA-Seq data to show the specificity of this compound, what else does it affect? Please comment. Nevertheless the data showing that VPS34, Aurora B and Survivin can rescue the micronuclei phenotype produced by TRABID depletion support the hypothesis, Fig 4c.

Ans: The TRABID inhibitor (TRABIDi) NSC 112200 was published in *BMC Chemical Biology* 12: 4, 2012. In this paper, the authors did show a dose-dependent inhibition against full-length TRABID (Fig. 3A, B; noted that the Flag-TRABID used in this assay is full length, see Methods for description) and IC50 is around 3 μ M. Furthermore, they did show a selectivity of TRABIDi towards TRABID, as inhibition was not observed with a 10-fold higher dose (30 μ M) on another OTU-family DUB, A20 (Fig. 3C, D). Nevertheless, the reviewer rightly pointed out that the activity towards K29 chain has not been tested. Therefore, we provided new data showing that

TRABIDi reverts the effect of TRABID on the K29-deubiquitination of Survivin (Supplementary Fig. 7a, left panel). However, TRABIDi has no effect on A20-mediated deubiquitination of ATG9A (Supplementary Fig. 7a, right panel). These data not only confirm the selectivity of TRABIDi on TRABID but also demonstrate an inhibitory effect of TRABIDi on TRABID-mediated K29-deubiquitination. Furthermore, we showed that the effect of TRABIDi on the induction of type I interferon responses is reversed by re-expression of TRABID substrates VPS34/Aurora B/Survivin (Fig. 5n), supporting the on-target effect of TRABIDi. Since the induction of type I interferon responses are important for anti-tumor immunity, data in Fig. 5n further support the on-target effect of TRABIDi on reshaping an anti-tumor immune microenvironment (data in Fig. 7). Finally, we provided new data showing that TRABIDi cannot further augment the micronuclei phenotype seen in Trabid KO MEFs (Supplementary Fig. 7b), which, we believe, represents strong evidence for the on-target effect of TRABIDi on micronuclei induction. Thus, with the multiple lines of evidence, we believe that the effects of TRABIDi described in this manuscript are unlikely caused by nonspecific activities. Nevertheless, we do agree with the reviewer for the importance of further characterizing the specificity, safety, and adverse effects of TRABIDi, including the utilization of omics-based strategies. However, such analyses are more suitable for a study that is centered on TRABID inhibitor discovery and is beyond the scope of current manuscript. We therefore chose to discuss these issues in the Discussion section (Lines 505-506).

The recent evidence that cGAS acts as a receptor to sense micronuclei and trigger micronucleophagy as well as cGAS autophagic degradation is a very interesting narrative to put the data obtained by the authors in context (Fig 4 and 5). The author rightfully explored the possibility that the reduced micronucleophagy observed upon TRABID depletion might stabilise cGAS. However, I find it difficult to reach the same conclusion given the overexposure of the GAPDH and the weak reduction in cGAS observed in Fig 5A. This data is not convincing enough to make the claim that TRABID also impacts on cGAS protein levels, in the absence of a rescue experiment. I also do not see an increase in p-STING in Fig 5E, given that the change observed for p-STING in matched in the FLAG WB (which seems overexposed). However, TRABID depletion seems to clearly have an effect on type I interferon genes and ISGs mRNA expression, although the authors should provide a clearer and more details explanation to try and reconcile this with the data from Jin J et al Nat Immunology who showed where IL12 and 23 levels were reduced upon TRABID deletion in mice. Please explain.

Ans: (1) We believe that the modest decrease of cGAS level seen in Fig. 5a is due to the existence of both negative and positive effects of micronuclei on cGAS abundance. On one hand, cGAS is a micronucleophagy receptor and therefore is degraded in lysosomes with micronuclei. However, as pointed out by reviewer #4 (see point 7 in p.24), cGAS is itself an ISG and therefore micronuclei-induced cGAS/STING pathway would lead to its upregulation. Thus, while this reviewer criticizes the modest decrease of cGAS in Fig. 5a, reviewer #4 “*is surprising to see AurkB knockdown cause substantial cGAS reduction*”. To reconcile the opinions of the two reviewers, we decided to use a more direct approach, that is, lyso-IP, to monitor the lysosome accumulation of cGAS. As shown in Fig. 5d, Aurora B knockdown leads to a marked increase of cGAS in lysosomes, whereas TRABID knockdown does not show an effect. Thus, these data support that TRABID deficiency, despite inducing micronuclei, could not trigger cGAS lysosomal accumulation due to the impairment of autophagy (micronucleophagy) activity. This notion is further validated by showing that re-expression of VPS34, the autophagic target of TRABID, in TRABID knockdown cells increases lysosomal recruitment of cGAS (Supplementary Fig. 5b). Thus, by combining the data of cGAS lysosome recruitment (lyso-IP) and cGAS degradation (Western blot), we believe that our conclusion is greatly strengthened. (2) As to the critique that “*it is not convincing enough to make the claim that TRABID also impacts on cGAS protein levels, in the absence of a rescue experiment.*” We would like to point out that we did perform rescue experiment showing that restoration of VPS34 expression in TRABID knockdown cells or TRABIDi-treated cells decreased cGAS levels (Fig. 5b, c). In revised manuscript, we further used lyso-IP to show that restoration of VPS34 expression in TRABID knockdown cells leads to an increase in cGAS recruitment to lysosomes (Supplementary Fig. 5b). (3) As to the p-STING data, we do not agree with the reviewer that “*change observed for p-STING is matched in the FLAG WB*”. In our original data, p-STING was highly induced by TRABID knockdown, whereas Flag WB (total STING) showed no change. Nevertheless, we do agree with the reviewer that the Flag WB is overexposed. Therefore, we repeated the experiment and provided new data in Fig. 5f, in which Flag WB is not overexposed while p-STING induction remains. In addition, to further convince this reviewer, we provided one more piece of evidence, that is, the induction of p-STAT1 by TRABID deficiency (Supplementary Fig. 5e), which represents the activation of interferon pathways. (3) As requested by the reviewer, we compared our findings with the paper by Jin et al. in the Discussion (Lines 496-506). Basically, the finding by Jin et al. is specific to dendritic cells, not T cells, and occurs specifically under the experimental autoimmune encephalomyelitis conditions, that is, mice are immunized with a peptide derived from myelin

oligodendrocyte glycoprotein together with pertussis toxin to trigger the severe autoimmune disease. However, under unstimulated conditions, TRABID does not affect DCs development, migration, and maturation. We would like to point out that the context used by the previous study is very different from that of tumor microenvironment (our study), which often displays immune suppressive features. Furthermore, it is unclear why TRABID-induced regulation of IL-12/IL-23 occurs only in DCs, but this might be due to the specific expression pattern of TRABID target (jmd2d) or the existence of additional regulatory mechanism. Based on the considerations from all these aspects, we do not see any conflict between our and the previous study, since the experimental conditions and cell contexts are completely different.

Although the transcriptomics data looks exciting, the literature has established a low correlation between mRNA expression and protein translation. Have the authors measured proteins levels of some of the IFNs, in particular PD-L1, for example by ELISA. This seems essential to make a more direct link here between the increase in PD-L1 upon TRABID depletion and the increase response to ICB.

Ans: We did provide ELISA data for the induction of secreted interferon β by TRABID knockdown (Fig. 5i). Furthermore, in revised manuscript, we included additional data showing that TRABID knockdown in HeLa and B16F10 cells induces PD-1 protein expression (Fig. 5m and Supplementary Fig. 5j).

The effect of TRABID depletion on reducing tumour growth and increasing an anti-tumor microenvironment is also convincing, but again a rescue experiment should be included to establish direct implication of TRABID DUB activity (Fig 6).

Ans: In Fig. 6d, e and Supplementary Fig. 6c of revised manuscript, we provided such rescue data. Of note, comparing with TRABID Δ N/CS mutant, TRABID wild type re-expression in TRABID knockdown cells leads to an increased tumor growth.

Figure 7 aims to establish that loss of TRABID function could improve efficacy of ICB through upregulation of PD-L1. This experiment set up by the authors is the correct one to test their hypothesis but I must reiterate my concerns with regards to the poorly characterised TRABID inhibitor used. This is in my mind is a major concern as there is not enough validation of this inhibitor, even in the original paper referenced, to bring confidence that this compound specifically target TRABID and not any other OTU DUB or the possibility that it could induce pleiotropic effect. The data look promising and unfortunately at this stage I am not aware of another publicly available TRABID inhibitor.

Ans: The critiques on TRABIDi was explained earlier. First, we clarified that the original paper did show a selectivity of this inhibitor on TRABID over A20, another OTU-family DUB. Second, we confirmed this selectivity by comparing the effect of TRABIDi for inhibiting TRABID's action on its target Survivin but fails to affect A20 on its substrate ATG9A (Supplementary Fig. 7a). In the same experiment, we also demonstrated the capability of TRABIDi to antagonize K29-ubiquitination. Third, we did show that the effect of TRABIDi on micronuclei and IFNs/ISGs induction are rescued by VPS34/Aurora B/Survivin co-expression (Fig. 5n). Of note, among the TRABIDi-induced genes that are validated by VPS34/Aurora B/Survivin co-expression, IFNB and IFNA14 can induce PD-L1 expression, whereas CXCL10 and CXCL11 are important chemokines for promoting the infiltration of anti-tumor immune cells. Based on current knowledge, the upregulation of these set of genes could contribute to the sensitization of tumor cells to ICB. Thus, even though we do not perform a rescue experiment to validate the effect of TRABIDi on ICB sensitization, the gene expression data do support an on-target effect of TRABIDi. Fourth, we showed that TRABIDi cannot further enhance the micronuclei phenotype seen in Trabid KO MEFs (Supplementary Fig. 7b), further supporting the on-target effect of TRABIDi. Finally, as pointed out by the reviewer, there is no other publicly available TRABID inhibitor. We thus consider that we have done our best to demonstrate the selectivity and on-target effects of TRABIDi. Nevertheless, there is indeed a need to further characterize this TRABIDi, which is included in the Discussion section (Lines 505-506).

It is not clear to me whether the authors are proposing that TRABID depletion reduces cGAS during mitosis or another cell cycle stage. Please comment.

Ans: We've never proposed that TRABID depletion reduces cGAS. In contrast, TRABID depletion activates cGAS/STING pathway and prevents cGAS from degradation by lysosomes. Regarding to the cell cycle position for the occurrence of each event shown in the model of Fig. 8d, the followings are our views based on current knowledge on cGAS/STING pathway and autophagy. First, it is well-documented that mitotic defects could lead to the generation of cytoplasmic micronuclei in the interphase (*Nature* 548: 466-470, 2017). That is, when cells exit from mitosis, nuclear envelop reforms at the outside of bulk chromosomes. However, if certain chromosomes fail to properly segregate during mitosis, they are left behind and cannot be incorporated into the newly formed nucleus. This leads to the generation micronuclei in the cytoplasm, which in turn induces the activation of cGAS/STING pathway. Of note, when cell cycle transits to the next mitosis, the breakdown of nuclear envelope facilitates the tethering of cGAS to chromatins, where

cGAS is inactivated by hyperphosphorylation (*Science* 371: eabc5386, 2021; also see point 9 of Reviewer #4). Similarly, the activities of autophagy are suppressed in mitosis (*Autophagy* 15: 1917-1934, 2019; *Mol Cell* 77:228-240, 2020). Thus, micronuclei appearance, activation of cGAS/STING pathway and autophagic degradation of micronuclei and cGAS all occur in the interphase even though the initial action point of TRABID is in M phase. We added one paragraph to discuss these points (Lines 552-563).

There is a lot of effects suggested to be driven by TRABID depletion and the narrative of the authors aims to place these different functions into a logical, linear pathway. Although I agree with the effect of TRABID depletion and the interaction with CPC, I am less convinced by the function of TRABID DUB activity in that process given the ubiquitination assay which I see as inconclusive and also the use of the TRABID inhibitor which has not been fully evaluated and validated. The effect of TRABID depletion on micronuclei and the induction of IFNs and ISGC mRNA transcription is also convincing but it is rather difficult to confidently put all these cellular processes into a linear pathway given the data provided.

Ans: (1) Regarding to the requirement of DUB activity of TRABID for the various effects described in the manuscript, we have provided new data indicating that the effects of TRABID on mitotic cell division (Fig. 1j), regulation of Aurora B and Survivin ubiquitination and abundance (Fig. 2f, 3a, 3b), suppressing micronuclei (Supplementary Fig. 4a) and tumor promotion (Fig. 6d, e and Supplementary Fig. 6c) are all dependent on its catalytic activity. In addition, we have significantly improved and strengthened the ubiquitination analyses. (2) Regarding to the concern on TRABIDⁱ, we have fully replied this in previous comments and therefore do not reiterate here. (3) Regarding to whether different effects of TRABID are mediated by a linear pathway, we first would like to point out that we do not propose that TRABID deficiency-induced anti-tumor immunity is mediated by a linear pathway. As illustrated in Fig. 8d, TRABID deficiency results in two different effects, i.e., mitosis defect and autophagy defect, through different sets of targets. It is the combination of these two effects that results in a robust activation of cGAS/STING-mediated anti-tumor immunity. In other word, TRABID deficiency elicits a bifurcated pathway to lead to the final outcome. Second, we did perform a large amount of rescue experiments to strengthen the causal relationships among the different effects seen in TRABID deficiency. First, we showed that re-expression of single set of TRABID targets partially rescues the effects of TRABID deficiency on micronuclei induction and cGAS/STING pathway target expression, whereas re-expression of both sets of targets completely rescues these effects (Fig. 4d and Fig. 5n). Second, the anti-tumor

responses of TRABID deficiency are reversed by STING inhibitor (Fig. 6p, q and Supplementary Fig. 6j) and attenuated in an immune-deficient microenvironment (Fig. 6f, g and Supplementary Fig. 6f). Third, the type I interferon responses of TRABID deficiency are abrogated by cGAS or STING inhibition (Supplementary Fig. 5k and 5l). Thus, if we view the entire pathway as a five-step process (1. TRABID deficiency; 2. VPS34 complex and CPC destabilization-induced autophagy and mitosis defects, respectively; 3. Micronuclei induction, 4. cGAS/STING activation, and 5. anti-tumor immunity), we did show that both step 3 and step 4 are inhibited by the blockage of step 2. Additionally, step 5 is inhibited by the blockage of step 4. Furthermore, based on our previous study (*Nature Commun* 12: 1322, 2021) and this revised manuscript, the links between step 1 and step 2 are rather strong, that is, VPS34 and Aurora B/Survivin as direct and physiological substrates of TRABID. Thus, we believe that the causal relationships of these steps have been rigorously studied. Having said this, we do not exclude the possibility for the existence of additional TRABID substrates that act together with VPS34/Aurora B/Survivin for similar effects on these steps and therefore have mentioned this in the Discussion (Lines 468-472).

Reviewer #2 (Remarks to the Author): with expertise in chromosomal instability, mitosis, cancer, STING

In their manuscript ‘TRABID inhibition activates cGAS/STING-mediated anti-tumor immunity through mitosis and autophagy dysregulation’, Chen and co-workers have studied the effects of depletion of inhibition of TRABID, more commonly referred to as ZRANB1.

The authors show a number of interesting phenotypes, and show that TRABID can act as a DUB for aurora B and survivin. Also, they show that inactivation of TRABID leads to inflammatory signaling, and sensitivity of trabid inhibited cells to ICI agents.

Overall, some of the phenotypes are impressive, but I feel that the observations remain poorly understood at the mechanistic level.

Ans: As can be seen in the revised manuscript, we have thoroughly addressed the critiques raised by each reviewer. As a result, the mechanistic insights have been greatly clarified and strengthened.

Specific comments:

- Line 6: ‘Since the DNA-sensing ability of cGAS is independent of DNA source and sequence, self DNAs accumulated in the cytoplasm enable the activation of cGAS/STING pathway’. Increasingly, the structure of extranuclear DNA is recognized to determine its ability to trigger cGAS. Actually, the chromatin context determines cgas activation, for instance PMID: 32913000

Ans: We added this aspect and the reviewer’s suggested reference to the sentence (Lines 73-74), together with a recently published *Nature Commun* article (PMID: 36732527), which fits the context even better.

-mitotic exit is poorly defined. In the upper example of the KO cells, there is no anaphase, so anaphase onset cannot be used.

(1) The readout of mitotic exit can vary with the experimental systems used. Since we used time-lapse microscopy to track cell behaviors, cell re-attachment was used as a definition of mitotic exit, which is explicated in the legends of Fig. 1b and Supplementary Fig. 1f. (2) We did not use the term of “anaphase onset” throughout the manuscript. The upper example of the *Trabid* KO cells in Fig. 1a exhibits a chromosome mis-segregation defect, which prohibits the identification of anaphase onset using image analysis.

-Is it surprising that 1 in 5 control cells does not have a bipolar spindle. I realize that

cells in culture have a background level of chromosome missegrgeations, but this is unexpectedly high.

Ans: The control and Trabid KO MEFs have gone through the spontaneous immortalization process. We do not realize whether this leads to an increase of the background. Of note, the control MEFs did show a higher mitotic error rate comparing to the control HeLa cells (comparing Fig. 1f with Fig. 1h, i). Considering the modest effect of Trabid KO on bipolar spindle and the unusually high background of the control MEFs (as pointed out by the reviewer), we decided to delete this piece of data in revised manuscript.

-to link the phenotype of increased micronuclei to an underlying mechanism, it is very important to test whether they are centromere positive

Ans: We showed in Fig. 4c of revised manuscript that centromeres are present in the micronuclei. This indicates that the micronuclei are generated by an aneugenic mechanism, consistent with the mitosis defects observed in TRABID deficient cells.

-Figure 2B: the color coding is difficult to interpret. Please use bargraphs or another way of displaying that is more quantitatively interpretable.

Ans: In Fig. 2b of revised manuscript, we changed the color coded figure into bar graph to show the fold changes and P values, as requested by the reviewer.

-Figure 2A/B: trabid does many things (as the authors also discuss in their introduction), including regulating transcription factors that are involved in proliferation. All the highlighted proteins are cell cycle regulated. Can a GSEA-type approach be used on the MS dataset to see if these proteins are part of a proliferation signature?

Ans: (1) The three highlighted proteins are all CPC components. In fact, it is conceivable that most proteins listed in Fig. 2a are cell cycle regulated, because they were either selected from the “Mitotic cell cycle” category of combined GO and Reactome analyses (Supplementary Fig. 2a) or brought out by their identity as CPC components (See Lines 167-175 for the description on the identification of mitotic regulators among the downregulated proteins by TRABID depletion). Of note, many cell cycle regulators are themselves regulated by cell cycle to elicit effects on a particular cell cycle period. (2) As requested by the reviewer, we performed GSEA analysis with the 441 downregulated proteins. Although statistically significant enrichments were obtained for p53 pathway, G2/M checkpoint, and mitotic spindle, each category contains only few proteins (see Fig. 1 in the next page). Thus, there is no evidence that TRABID could regulate certain transcription factor to control

mitosis. (3) We do not believe that TRABID-induced stabilization of CPC subunits is secondary to a cell cycle regulatory effect of TRABID. We showed in Fig. 3g-i that purified TRABID directly binds and deubiquitinates Aurora B and Survivin in vitro. Furthermore, while TRABID does not affect INCENP ubiquitination level, the downregulation of Aurora B or Survivin leads to the downregulation of INCENP (Supplementary Fig. 3a, 3g). These data provided strong evidence that Aurora B and Survivin are direct targets of TRABID, whereas INCENP is regulated indirectly by TRABID through the stabilization of Aurora B and Survivin.

Fig. 1. GESA plots for the match of downregulated proteins by TRABID knockdown with the indicated signatures.

Figure 2C/D: many of the observed effects can be explained by trabid-depleted cells to proliferate slower. This would be checked. Also, the treatments in Figure 2C/D should be accompanied by flow cytometry analysis (DNA, plus mitotic marker).

Ans: Fig. 2c, d showed an upregulation of TRABID in M phase, which was revealed by treating cells with agents that arrest cell cycle at different positions. We could not see any linkage of this finding to the effect of TRABID on cell proliferation. In addition, we did not analyze any effect of TRABID nor used TRABID-depleted cells in Fig. 2c, d. Regarding to the flow cytometry data for the cell cycle profiling of cells treated with different inhibitors, we provided such data in Supplementary Fig. 2b.

Figure 3: These data show that overexpressed trabid can deubiquitlate overexpressed survivin and aurora B in vitro, and that depletion of trabid leads to slightly elevated levels of ubiquitylated versions of overexpressed survivin and aurora B. the authors should also show that endogenous survivin or aurora B are more ubiquitylated in the absence of trabid.

Ans: In revised manuscript, we included new data showing that TRABID knockdown increases K29-ubiquitination of the endogenous Aurora B and Survivin (Fig. 3e, f). Furthermore, we repeated experiments in Fig. 3a-d by using a more stringent condition to reduce the background (as suggested by reviewer #1), which results in more robust effects on Aurora B and Survivin ubiquitination levels by TRABID overexpression or knockdown (Fig. 3a-d). Finally, by using a K29-ubiquitin chain-

specific binder (sAB-K29), we demonstrated the ability of TRABID to remove this Ub chain type from Aurora B and Survivin (Supplementary Fig. 3c-f). Thus, the function of TRABID on Aurora B and Survivin deubiquitination is greatly strengthened in revised manuscript.

Also, it appears that traid depletion can lead to survivin or aurora B ubiquitinylation in nocodazole, a situation when APC/C-Cdh1 is not active yet due to high CDK activity. This is highly unexpected. How do the authors explain these data?

Ans: We realized and explained this thoroughly in the Discussion (Lines 508-530). In brief, we do not believe that TRABID antagonizes the function of APC/C^{Cdh1} for two reasons: the unmatched action time points in the cell cycle (as pointed out by this reviewer) and, more importantly, the unmatched ubiquitin chain type specificities between APC/C and TRABID. Thus, TRABID likely counteracts the action of an Aurora B E3 ligase other than APC/C^{Cdh1}. There are indeed other E3 ligases that can target Aurora B for ubiquitin-dependent degradation (see Introduction, Lines 88-89). Of note, USP35, another DUB, also deubiquitinates and stabilizes Aurora B in M phase and USP35 is induced by FoxM1 in G2/M, prior to the action of APC/C^{Cdh1} (PMID: 29449677). Furthermore, USP13 was reported to target Aurora B for deubiquitination and stabilization in the interphase (PMID:32772043). These previous studies support the possibility to stabilize Aurora B through deubiquitination before the activation of APC/C^{Cdh1}.

Figure 4: This figure needs western blots and IF to confirm that the levels of aurora B, survivin and VPS34 are expressed to control levels. Also, transient transfection usually does not give a 100% transfection efficiency, although microscopy is used to analyse all cells? How do the authors know if all cells that are analyzed contain all the transiently expressed constructs? Same applies to figure 5i.

Ans: (1) We included Western blot data for Fig. 4d, 4e, and 5n in revised manuscript, which demonstrated the rescue of Aurora B, Survivin and VPS34 expression. (2) In our experiments, transfection efficiency was >95%, as monitored by GFP fluorescence. Furthermore, the percentage of cells positive with GFP fluorescence was not changed by cotransfection with all three constructs (GFP-Aurora B, surviving, and VPS34). Thus, most transfected cells likely contain all three constructs. Nevertheless, it is possible that a small percentage of cells that do not contain all three constructs and this might explain the inability of co-transfection of all three constructs to revert the micronuclei level down to that of control cells (comparing lanes 8 and 12 with lane 1 in Fig. 4d). However, this would not affect the conclusion of our studies, because we are able to show the differential effects of single/double and triple

transfections.

Figure 5a/b: TRabid aurora B and blots should be included to control for knockdown.

Ans: Stable cell lines are used in these two experiments. The TRABID knockdown data are shown in Fig. 2e, whereas Aurora B knockdown data are in Supplementary Fig. 3g. Such information is provided in the figure legends.

Figure 5: ‘..while Aurora B depletion reduced cGAS abundance through a lysosome-dependent mechanism’. This could indeed be, but there are so many controls missing here, that is a wild overinterpretation of the results.

Ans: The reason for lacking many controls is probably due to the utilization of an indirect approach, that is, Western blot analysis of cGAS level, to infer to the lysosomal degradation of cGAS. To strengthen the conclusion, we performed lyso-IP analysis to directly monitor lysosome recruitment of cGAS. In Fig. 5d revised manuscript, we found that Aurora B knockdown greatly increases lysosome accumulation of cGAS, whereas TRABID knockdown displays no effect. This suggests that TRABID has another target that compensates/antagonizes this effect of Aurora B. In Supplementary Fig. 5b, we showed that re-expression of VPS34, another TRABID target, in TRABID knockdown cells phenocopies the effect of Aurora B knockdown. Together, these data are consistent with the conclusion that TRABID knockdown-induced VPS34 downregulation impairs autophagy to prevent autophagy (micronucleophagy)-mediated degradation of cGAS. By combining the data of cGAS lysosome recruitment (lyso-IP) and cGAS downregulation/lysosome degradation (Western blot), we believe that our conclusion has been greatly strengthened.

Figure 5: the authors should show that the effects are dependent on cGAS/STING, through siRNA or knockouts of these genes.

Ans: We showed in Supplementary Fig. 5k, l of revised manuscript that cGAS inhibitor G140 or STING inhibitor C176 each blocks TRABID knockdown-induced expression of IFNB (an interferon) and CXCL10 (an ISG), thus demonstrating an dependence on cGAS/STING pathway.

Figure 6: the TRABID-depleted tumors are smaller and grow less fast. This could indicate strong defects in proliferation, that also affect all other assays in the manuscript. The authors should check for proliferation rates (ki67 positivity in tumors appears very different!).

Ans: Trabid knockdown did not significantly affect the proliferation of B16F10 cells (Supplementary Fig. 6e), which might be due to the inactivation of mitotic checkpoint

in this cancer cell line. To further address the immune dependency of the tumor suppressive effects of Travid knockdown seen in Fig. 6, we implanted the same set of control and Travid knockdown B16F10 cells into Nude mice, which lack T/B cells and with a reduced NK function. Travid knockdown only modestly decreases tumor formation in Nude mice. In view of the drastic difference in the Travid knockdown effect seen in immune-proficient mice (Fig. 6a, b and Supplementary Fig. 6a) and immune-deficient mice (Fig. 6f, g and Supplementary Fig. 6f), the tumor immune microenvironment should play a key role in the suppressive effect of Travid knockdown on tumor growth. The slight decrease of tumor growth seen in Nude mice by Travid knockdown could be attributed to the effect of macrophages (Travid in tumor does induce macrophage M2 polarization as shown in Fig. 6n, o). In sum, tumor cell proliferation *per se* should not play a major role in the tumor suppressive effects of Travid knockdown seen in Fig. 6. The drastic difference in Ki67 staining seen in Supplementary Fig. 6d is likely due to the tumor suppressive effect of infiltrated immune cells in animals inoculated with Travid knockdown cells.

The manuscript needs extensive grammar checks. Few examples below:

Line 29 prevents -> protects

Line 42: show -> shows

Line 44: of -> to

Line 48: 'for' should be left out

Line 60: 'are testing' -> have been tested?

Line 132: 'no' -> not

Ans: We apologize for these grammar errors and have thoroughly checked the manuscript to correct these and other mistakes.

Reviewer #3 (Remarks to the Author): with expertise in autophagy, cGAS/STING

In this study, Chen YH et al examined a novel function of TRABID, a DUB in regulating cGAS/STING-mediated anti-tumor immunity via altered mitosis and autophagy. First, they established the role of TRABID in mitosis by using TRABID-KO MEFs and other cells. Second, they found that TRABID is upregulated in mitosis and that TRABID regulates mitosis through deubiquitinating and stabilizing CPC such as Aurora B and Survivin. Third, TRABID deficiency causes micronuclei through a combinatory defect in mitosis and autophagy. Fourth, with the increase level of cytosolic micronuclei caused by TRABID deficiency, the cGAS/STING pathway is activated that contributes to the tumor-suppressive effect of Trabid deficiency. Fifth, using animal models, the authors provided evidence that inhibition of TRABID sensitizes tumor response to anti-PD-1 therapy. Finally, TRABID expression in most solid cancer types correlates inversely with an interferon signature and infiltration of anti-tumor immune cells. Overall, this is a very comprehensive study with convincing data, covering cell, animal and clinical aspects with deep mechanistical understanding and potential application in cancer immunotherapy.

Ans: We appreciate this reviewer for supporting our study.

There are several points to be addressed to improve the MS.

1) In the mechanistic study, the authors presented two parallel pathways: mitosis defects and impaired autophagy, without establishing the possible link btw these two processes. Based on the knowledge that autophagy is also implicated in regulation of cell cycle esp the M phase (PMID 31733992, 30898011), it would be important for the authors to address this possible link, or at least discuss this point.

Ans: In fact, the two papers are not related to the role of autophagy in regulating cell cycle or M phase. Instead, both indicated that autophagy activity is suppressed in M phase, albeit with different mechanisms. In the first paper (PMID: 31733992), the authors reported that the master M phase kinase CDK1 substitutes mTORC1 to phosphorylate and inactivate a number of key autophagic proteins, including ULK1, ATG13, ATG14, and TFEB, thereby resulting in the suppression of autophagy. In the second paper (PMID: 30898011), in which our lab. contributed partly to the study, the authors reported the activation of Cul4 E3 ligase through neddylation in M phase to result in the ubiquitination and degradation of autophagic protein WIPI2, thereby suppressing autophagy. Since the molecular mechanisms involved in the current study have no overlapping with the two previous studies, we cannot find any linkage

between our and previous studies. Furthermore, these previous studies cannot provide any clue to link the two effects of TRABID, that is, mitosis and autophagy, or the two sets of TRABID targets responsible for the two effects, that is, Aurora B/Survivin and VPS34. Finally, we do not see any problem for the function of TRABID to regulate the two important cellular processes through parallel pathways. In fact, it is the combination of the two processes that makes TRABID a potent target for activating anti-tumor immunity. Nevertheless, these previous studies suggest that the two different effects of TRABID occur at different phases of cell cycle. To point out this aspect and to reply the question raised by reviewer #1 (see p.11 of this letter), we added one paragraph in Discussion to explain the different timings of the various events elicited by TRABID deficiency (Lines 552-563), in which the two papers are cited.

2) In the in vitro experiments using cell lines, the authors only used siRNA knockdown of TRABID in HeLa cells. The study will be significantly enhanced if they can establish the TRABID KO stable cells using CRISPR (in addition to the MEFs), plus the reconstitution of both the WT and enzyme-activity dead mutant, to replicate some of the key observations such as changes of Aurora B and Survivin protein level, formation of cytosolic micronuclei, etc.

Ans: In revised manuscript, we provided data of rescue experiments (re-expressing TRABID WT and catalytically dead mutant in TRABID knockdown cells) for mitotic cell division (Fig. 1j), Aurora B/Survivin expression levels (Fig. 2f), micronuclei (Supplementary Fig. 4a), and effect on tumor growth in vivo (Fig. 6d, e and Supplementary Fig. 6c). In addition, we provided transient siRNA knockdown data for mitotic cell division (Fig. 1i). With the large amount of new data, we believe that we have confirmed the on-target effect of TRABID shRNAs and the dependence of TRABID catalytic activity on its functions identified in this study.

3) One key point of this study is that TRABID deubiquitinates and stabilizes Aurora B and Survivin. However, as stated by the authors, TRABID is a DUB that specifically disassembles K29- and K33-linked ubiquitin chains. Then, the authors need more evidence to highlight that how K29- and K33 Ub are implicated in regulating the protein stability of Aurora B and Survivin. Will MG132 prevent the reduction of protein levels of CPC components in shTRABID cells treated with nocodazole (Figure 2e)?

Ans: (1) We did show that that MG132 prevents the downregulation of Aurora B and Survivin by TRABID knockdown in Fig. 3k. (2) In revised manuscript, we provided solid evidence for the role of TRABID in removing the K29-linked ubiquitin chains

from Aurora B and Survivin in vivo (Fig. 3e, f and Supplementary Fig. 3c-f) and in vitro (Fig. 3h, i). It is known that the cellular K29 ubiquitin chains are mostly, if not entirely, presented as the K29/K48 branched chains (PMID: 25752573; PMID: 28291339). Furthermore, we and other uncovered that this chain type undergoes an enhanced proteasomal degradation (PMID: 33567268; PMID: 33637724). Thus, it is very likely that Aurora B and Survivin are modified by K29/K48 branched ubiquitin chains, leading to a proteasomal degradation fate, and TRABID antagonizes this ubiquitination to stabilize the two proteins.

Other points:

1) *How does nocodazole treatment (cells in mitosis) enhance the expression of TRABID (both mRNA and protein)?*

Ans: The only clue that we have so far is the upregulation of TRABID mRNA and protein levels by the active mutant of transcription factor FoxM1 (i.e., FoxM1 Δ N) (see Fig. 2 below), which is a master regulator of many G2/M phase genes. However, whether FoxM1 is required for nocodazole-induced TRABID expression and whether TRABID is a direct target of this transcription factor require further analyses. Given the comprehensive content of this study, we consider that this part may be more suitable for a separate study centered on the TRABID regulation.

Fig. 2. Western blot (left) and RT-qPCR (right) analyses of TRABID expression in HeLa cells transiently transfected with FoxM1 Δ N.

2) *Figure 8d: In this illustration, the link between cGAS and micronuclei is not clearly illustrated.*

Ans: We revised this model to reveal the link.

Reviewer #4 (Remarks to the Author): with expertise in cGAS/STING

This manuscript reports a series of previously unrecognized consequence of lacking the ubiquitin thioesterase, TRABID, aka ZRANB1, in cells. First, the authors show that TRABID deficiency caused chromatin mis-segregation and cytokinesis failure. Such mitotic errors lead to more micronuclei, thus activate the cGAS-STING pathway. Then they applied quantitative proteomics to show that a few proteins are downregulated upon TRABID knockdown, including Aurora Kinase B and Survivin, which play critical roles in mitotic chromatin segregation. Plus, they demonstrate that TRABID can de-ubiquitinate Aurora Kinase B and Survivin. Because TRABID deficiency engages the innate immune response, the authors went further to show that removing TRABID in tumor cells can elicit anti-tumor response and TRABID inhibitor can improve the anti-tumor effect of immune checkpoint blockade by the anti-PD-L1 antibody. Together, this is a quite comprehensive study that report some interesting findings on TRABID.

Ans: We appreciate the reviewer for supporting our study.

However, the authors must address the following major and minor concerns.

1. TRABID specifically bind and remove K29-linked and K33-linked ubiquitin. Are Aurora B and Survivin modified by these special forms of ubiquitin chains?

Ans: In revised manuscript, we used a K29-ubiquitin chain-specific binder (sAB-K29) to provide a large amount of data indicating not only that Aurora B and Survivin are modified by K29-linked polyubiquitin chains, but also that TRABID is capable of removing such ubiquitin chains from the two substrates in vivo (Fig. 3e, f and Supplementary Fig. 3c-f) and in vitro (Fig. 3h, i).

2. TRABID regulates many different proteins via de-ubiquitination, as the SILAC experiments show. How confident are you that the effect on cGAS dysregulation is mainly via Aurora Kinase B and VPS34? In those rescue experiments with Aurora B and VPS34 overexpression, it is necessary to show by immunoblotting that levels of these proteins are restored to normal levels.

Ans: (1) The induction of micronuclei and IFN/ISG genes by TRABID knockdown are rescued by overexpression of Aurora B/Survivin/VPS34 (Fig. 4d and 5n). Thus, we believe that these TRABID substrates play a major role in mediating TRABID's effects on micronuclei induction, cGAS/STING regulation and the induction of type I IFN responses. However, as described in the Discussion (Lines 468-472), we cannot rule out the involvement of other TRABID targets in the process. (2) Western blot data for the substrate rescue experiments are all included in the revised manuscript

(Fig. 4d, 4e, and 5n). The expression of these substrates are indeed restored to normal levels.

3. Page 13 Fig Sfig 5. *cGAS inhibitor G140 can reduce interferon in TRABID knockdown cells. It would be more convincing to show similar experiments with STING inhibitor H151/C176, and with specific knockout/knockdown of cGAS or STING.*

Ans: We provided new data showing that STING inhibitor C176 reverts the effect of TRABID knockdown on the expression of IFN and ISG (Supplementary Fig. 5l).

4. Fig 3a, second lane: *as there was no his-Ub, how come there was GFP-Aurora B in the IP? Fig 3b, what is the 70kDa band in Lane 2 of the anti-his immunoblot?*

Ans: We apologize for the poor quality of the deubiquitination assays shown in Fig. 3a, b. In revised manuscript, we repeated these experiments with more stringent wash conditions and eliminated the nonspecific bands.

5. Fig 3f-g showed that purified his-TRABID can de-ubiquitinate Aurora Kinase B and Survivin in vitro. *The concern is whether the activity comes from his-TRABID or co-purified contaminant(s). Please show Coomassie gel of his-TRABID to show protein purity. Also consider use catalytic dead mutant of TRABID as a negative control.*

Ans: The His-TRABID was obtained from a commercial source. Coomassie blue staining is provided in revised manuscript (Fig. 3h, lower panel) to show the purity. The catalytically dead mutant is not commercially available, thus precluding the possibility for a comparison.

6. Page 3 Line 53 “*cGAS binds and activates STING...*” *cGAS does not bind STING. It activates STING via its product, second messenger, cGAMP.*

Ans: We cannot find these words. In fact, our sentence is “cGAMP binds and activates STING...” (Lines 57-58), which should have no mistake.

7. Fig 5a, *why would Aurora Kinase B knockdown cause reduction in cGAS level via lysosomal-dependent mechanism? Aurora Kinase B knockdown will cause mitotic errors, like TRABID knockdown does, and it will likely induce cGAS activation via micronuclei. This may lead to increased lysosomal degradation of micronuclei. However, not all cGAS is in micronuclei, and elevated interferon may also induce more cGAS (as an ISG). Thus, it is surprising that AurkB knockdown cause substantial cGAS reduction.*

Ans: A recent study identified cGAS is itself an autophagic receptor for the autophagic degradation of micronuclei (*Autophagy* 17: 3976-3991, 2021). Thus, while cGAS is activated by cytoplasmic micronuclei, it also undergoes lysosomal degradation in response to micronuclei (Fig. 6A of this paper). It is rightly pointed out by the reviewer that not all cGAS is in the micronuclei and, furthermore, cGAS is itself an ISG and therefore is induced upon the activation of cGAS pathway. Due to both positive and negative impacts of micronuclei on cGAS abundance, the net change at its protein level is modest and different reviewers have different expectations on the extent of cGAS level changes. While this reviewer is surprising to see the substantial reduction, reviewer #1 considered that “the change is too modest and therefore is not convincing” (see p. 8 of this letter). To reconcile the opinions of the two reviewers, we decided to directly monitor lysosomal recruitment of cGAS by lyso-IP, thus avoiding the caveat generated by the bi-directional regulations of cGAS. As shown in Fig. 5d, Aurora B knockdown leads to a marked increase of cGAS in lysosomes, whereas TRABID knockdown does not show any effect. These data suggest that TRABID deficiency, despite inducing micronuclei, could not trigger cGAS lysosomal degradation due to the impairment of autophagy activity. This notion is further validated by showing that re-expression of VPS34, the autophagic target of TRABID, in TRABID knockdown cells increases cGAS recruitment to lysosomes (Supplementary Fig. 5b). With the addition of lyso-IP data, we believe that we have provided solid evidence for the role of TRABID knockdown in preventing cGAS from micronucleophagy-induced degradation through the impairment of autophagy activity.

8. Fig 6a. TRABID knockdown retarded tumor growth of B16F10 cells. One possible explanation is that TRABID knockdown may reduce the fitness of B16F10 through compromising cell cycle progression, which by itself would be sufficient to reduce tumorigenesis. How much does the anti-tumor immunity contribute to this reduced tumorigenesis is unknown.

Ans: To address this issue, we inoculated the same set of B16F10 cells to Nude mice, which lack T/B cells and with a reduced NK activity. Our data showed that Travid knockdown only slightly decreases tumor growth in Nude mice. In viewing of the drastic difference in the Travid knockdown effect seen in immune-proficient mice (Fig. 6a, b and Supplementary Fig. 6a) and immune-deficient mice (Fig. 6f, g and Supplementary Fig. 6f), the tumor immune microenvironment system should play a key role in the suppressive effect of TRABID knockdown on tumor growth. The slight decrease of tumor growth seen in Nude mice by TRABID knockdown could be attributed to the effect of macrophage (TRABID in tumor does induce M2

macrophage polarization as shown in Fig. 6n, o), tumor cells *per se*, or the combination of both. Nevertheless, we showed that Trabid deficiency does not significantly reduce cell proliferation in the cell culture system (Supplementary Fig. 6e), perhaps due to a defect in mitotic checkpoint in this cancer cell line. In sum, our data indicate that tumor cell proliferation should not play a major role in the tumor-suppressive effect of TRABID knockdown seen in Fig. 6a, b.

9. Aurora Kinase B was recently reported to phosphorylate cGAS during cell cycle, and hyperphosphorylation inhibits cGAS activity. Conceptually, this connection might play a role in the immune outcome of TRABID deficiency. Should include this in the discussion.

Ans: We appreciate this valuable suggestion and added this aspect to the Discussion (Lines 555-559).

REVIEWERS' COMMENTS

Reviewer #1 (Remarks to the Author):

I congratulate the authors for their comprehensive response including following up on some of the suggestions made including using the sAB-K29 as well as rescue experiments with wild type and mutant TRABID. The revised manuscript is much improved and supported by the new data provided.

The new data included confirms TRABID DUB activity is required to stabilise AuroraB and Survivin protein levels (fig 2f); new IP data further validate the interaction between TRABID and AuroraB, Borealin and Survivin (Fig 2h); the use of the recently developed sAB-K29 aptamer shows AuroraB and survivin are modified by K29 ubiquitin chains (Fig 3 and Supp fig); TRABID depletion increases p-STING and PD-L1 levels. The authors also used an alternative approach, lyso-IP, to specifically show TRABID depletion leads to cGAS lysosomal turnover, and this is a compelling piece of data (Fig 5d).

The rescue assays with wild type or a mutant TRABID construct which cannot bind nor cleave ubiquitin chains were attempted in various assays and the results further validate the original observations made included that the phenotypes observed can be directly attributable to TRABID.

My concerns regarding the TRABIDi remains but I do appreciate and accept the authors 's response. The authors have used the only compound available and have done their best to provide new data. In particular the new data presented in Supp Fig 7A would support the effect of TRABIDi in inhibiting TRABID-mediated K29 deubiquitination which has not been shown before. I agree with the authors that the characterisation of this inhibitor is beyond the scope of their work and in the absence of any alternative reagents to inhibit TRABID activity the data provided seems therefore adequate and the best one could do at this stage.

All comments I have made with regards to the context/hypothesis/narrative/clarifications have been addressed in the main text and/or through justification in the rebuttal.

Some of the new data provided would have benefited from having the control and test experiments undertaken at the same time and showed on the same figure. This is the case for example for Fig1 I and J which should really have been done as part of the same experiment rather than two, as suggested by the different figures. Same comment for Fig 6D. As part of this experiment, there should have been an shRNA alone or shRNA + Ev condition alongside the TRABID WT vs Mutant rescues on the same figure. This is unlikely to affect the validity of the data but it is not ideal. I am not suggesting redoing these experiments.

There are no controls to show bafilomycin is working in this assay (Fig 5A).

Data for Fig 4D is almost exactly the same as the one for Fig5N. I am sceptical with regards to the data suggesting the TRABID inhibitor phenocopies the shRNA data to the same extent. The effect looks identical which is rather difficult to believe in the absence of an appropriate control (Fig 5N). To address this there should be a WB showing endogenous TRABID protein levels for both figures.

Some WB were VPS34 has been overexpressed show a signal intensity similar to the non-transfected lane which is surprising. For these figures, there should be a WB showing endogenous proteins levels and another showing overexpressed protein levels (Fig 4C, D, 5B, 5N). This is also true for WB of TRABID in Supp Fig 4a. A blot showing detection of endogenous TRABID vs overexpressed would be more appropriate.

I must have missed this in the manuscript but what is preventing non-cancerous cells from being targeted by the proposed mechanism. Presumably TRABID depletion will induce mitotic defect and cGAS/STING activation, including increase PD-L1, in all cell types including non-cancerous cells?

Reviewer #2 (Remarks to the Author):

The authors have extensively addressed my comments, I support publication

Reviewer #3 (Remarks to the Author):

In this revised manuscript, the authors have carefully addressed most of the comments from the 4 reviewers, including mine. The new data and revisions have improved the quality of the whole study. I have no further comments.

Reviewer #4 (Remarks to the Author):

The revised manuscript has largely addressed my initial concerns.

Response to reviewer #1's comment

Reviewer's Comments:

Reviewer #1 (Remarks to the Author)

I congratulate the authors for their comprehensive response including following up on some of the suggestions made including using the sAB-K29 as well as rescue experiments with wild type and mutant TRABID. The revised manuscript is much improved and supported by the new data provided.

The new data included confirms TRABID DUB activity is required to stabilise AuroraB and Survivin protein levels (fig 2f); new IP data further validate the interaction between TRABID and AuroraB, Borealin and Survivin (Fig 2h); the use of the recently developed sAB-K29 aptamer shows AuroraB and survivin are modified by K29 ubiquitin chains (Fig 3 and Supp fig); TRABID depletion increases p-STING and PD-L1 levels. The authors also used an alternative approach, lyso-IP, to specifically show TRABID depletion leads to cGAS lysosomal turnover, and this is a compelling piece of data (Fig 5d).

The rescue assays with wild type or a mutant TRABID construct which cannot bind nor cleave ubiquitin chains were attempted in various assays and the results further validate the original observations made included that the phenotypes observed can be directly attributable to TRABID.

Ans: We appreciate the support of this reviewer.

My concerns regarding the TRABIDi remains but I do appreciate and accept the authors 's response. The authors have used the only compound available and have done their best to provide new data. In particular the new data presented in Supp Fig 7A would support the effect of TRABIDi in inhibiting TRABID-mediated K29 deubiquitination which has not been shown before. I agree with the authors that the characterisation of this inhibitor is beyond the scope of their work and in the absence of any alternative reagents to inhibit TRABID activity the data provided seems therefore adequate and the best one could do at this stage.

Ans: We appreciate the understanding of this reviewer.

All comments I have made with regards to the context/hypothesis/narrative/clarifications have been addressed in the main text and/or through justification in the rebuttal.

Ans: We appreciate the support of this reviewer.

Some of the new data provided would have benefited from having the control and test experiments undertaken at the same time and showed on the same figure. This is the case for example for Fig1 I and J which should really have been done as part of the same experiment rather than two, as suggested by the different figures. Same comment for Fig 6D. As part of this experiment, there should have been an shRNA alone or shRNA + Ev condition alongside the TRABID WT vs Mutant rescues on the same figure. This is unlikely to affect the validity of the data but it is not ideal. I am not suggesting redoing these experiments.

Ans: We agree with the reviewer that combination of the knockdown and rescue experiments would be more ideal. Nevertheless, there are reasons for the separate experiments. For Fig. 6, the knockdown experiments (Fig. 6a-c) were presented in the original manuscript, whereas the rescue experiment (Fig. 6d) was requested by the reviewer. To minimize the number of animals used in the study (following the 3Rs principle of animal experiments), we did not repeat the knockdown experiment when we conducted the rescue experiment. For Fig. 1i, j, both experiments were conducted in revision. However, analysis of mitotic phenotypes and establishment of new stable cell lines are both time consuming. Given the short revision period, we chose to conduct the transient knockdown experiment first (Fig. 1i). After the establishment of stable cell line, we performed mitosis analysis for the rescue experiment (Fig. 1j).

There are no controls to show bafilomycin is working in this assay (Fig 5A).

Ans: Bafilomycin treatment did show an effect in Fig. 5a, that is, blockage of cGAS degradation. Thus, the data by themselves indicate the validity of bafilomycin treatment. There is no need to include a control. Similarly, bafilomycin treatment in Fig. 4e (with the same treatment conditions as Fig. 5a) also showed an effect on micronuclei. Finally, Fig. 5a and Fig. 4e were both presented in the original manuscript and no control was requested by the reviewer previously.

Data for Fig 4D is almost exactly the same as the one for Fig5N. I am sceptical with regards to the data suggesting the TRABID inhibitor phenocopies the shRNA data to the same extent. The effect looks identical which is rather difficult to believe in the absence of an appropriate control (Fig 5N). To address this there should be a WB showing endogenous TRABID protein levels for both figures.

Ans: We guess the reviewer meant the Western blot data. (1) We do not see any problem for the similarity between the two sets of data because they represent similar experiments except for the replacement of TRABID knockdown with TRABID

inhibition in Fig. 5n. In all other experiments shown in this manuscript involving TRABIDi, the effects are similar to TRABID knockdown. This suggests not only an efficient blockage of TRABID activity by TRABIDi but also the consistency of our findings. (2) We do not understand the critique for the absence of an appropriate control for Fig. 5n. The Western blot data represent the protein expression controls for RT-qPCR data shown in Fig. 5n or micronuclei data shown in Fig. 4d. We do not know what kind of control is needed for these control experiments. Inclusion of data for TRABID protein level, as suggested by the reviewer, would not be meaningful because the two experiments would not show the same results. In Fig. 4d, TRABID expression would be reduced in the group with TRABID shRNAs, whereas TRABID level would not differ in any group of Fig. 5n. This is because TRABIDi blocks TRABID activity without changing its expression level (See Supplementary Fig. 7a, the V5 blot). Thus, the effects on Fig. 4d and 5n would not correlate with TRABID expression levels.

Some WB were VPS34 has been overexpressed show a signal intensity similar to the non-transfected lane which is surprising. For these figures, there should be a WB showing endogenous proteins levels and another showing overexpressed protein levels (Fig 4C, D, 5B, 5N). This is also true for WB of TRABID in Supp Fig 4a. A blot showing detection of endogenous TRABID vs overexpressed would be more appropriate.

Ans: (1) Fig 4c showed FISH data of centromeres. No Western blot is involved. (2) Regarding to the VPS34 WB data in Fig. 4d, 5b, and 5n, we do not understand why the reviewer considered surprising for the similar levels observed from rescued and control lanes. Of note, these control experiments were requested by Reviewer 2 and 4. The former asked us to “confirm that the levels of Aurora B, Survivin and VPS34 are expressed to control levels”, whereas Reviewer 4 indicated that “it is necessary to show by immunoblotting that levels of these proteins are restored to normal levels”. Thus, restoration to the normal/control levels is what expected/requested by the two reviewers. One possibility is that this reviewer actually meant that the levels of VPS34 in the rescue experiments (lanes 4, 5) are the same as that in control cells (lane 1). However, the levels of Aurora B and Survivin in lane 4, 5 are slightly higher than lane 1. There are two possibilities for the differences. First, the rescued amount for VPS34 is lower than Aurora B and Survivin. Second, the antibody for VPS34 might be less sensitive to reveal expression differences comparing to antibodies against Aurora B and Survivin. Thus, the range of intensity differences seen in the blot with VPS34 antibody is smaller than the blots with other antibodies. Nevertheless, neither scenario would affect the conclusion as the control data did show the downregulation

of VPS34 by TRABID shRNA and the rescue effect by VPS34 re-expression. (3) Regarding to the comment for separate blots showing endogenous and overexpressed protein levels, respectively, we would like to point out that detection of endogenous protein only is impossible for this experiment, as antibody recognizing the endogenous protein would also react with overexpressed proteins. We do not know whether the reviewer actually meant the inclusion of a separate blot using antibodies to the tags of overexpressed proteins. If so, we do not know the purpose of this blot, as this piece of data can only indicate that the transfected proteins are expressed. This conclusion has already been reached by our current data. That is, a rescue effect should indicate the expression of transfected proteins. Similar arguments apply to Supplementary Fig. 4a.

I must have missed this in the manuscript but what is preventing non-cancerous cells from being targeted by the proposed mechanism. Presumably TRABID depletion will induce mitotic defect and cGAS/STING activation, including increase PD-L1, in all cell types including non-cancerous cells?

Ans: We did discuss this in the Discussion section (Lines 491-496). The sentences are copied below. “Of note, a previous study indicated that germ-line deletion of Zranb1 shows no or minor effects on the development of various cells in the immune system⁶¹, suggesting that Trabid may not be an indispensable factor for the mitosis of immune cells. Perhaps other DUBs expressed in these cells could compensate Trabid loss for stabilizing CPC.”